ecology

bearded vulture, risk mitigation, wildlife–human conflicts, spatial planning, predictive modelling, vulture conservation

# A predictive flight-altitude model for avoiding future conflicts between an emblematic raptor and wind energy development in the Swiss Alps

Sergio Vignali[1], Franziska Lörcher[2,3,4], Daniel Hegglin[2,3], Raphaël Arlettaz[1] and Veronika Braunisch[1,5]

[1]Division of Conservation Biology, Institute of Ecology and Evolution, University of Bern, Bern, Switzerland
[2]Stiftung Pro Bartgeier, [3]SWILD, and [4]Vulture Conservation Foundation, Wuhrstrasse 12, 8003 Zurich, Switzerland
[5]Forest Research Institute of Baden-Wuerttemberg, Wonnhaldestrasse 4, 79100 Freiburg, Germany

SV, 0000-0002-3390-5442; RA, 0000-0001-6360-5339; VB, 0000-0001-7035-4662

Deployment of wind energy is proposed as a mechanism to reduce greenhouse gas emissions. Yet, wind energy and large birds, notably soaring raptors, both depend on suitable wind conditions. Conflicts in airspace use may thus arise due to the risks of collisions of birds with the blades of wind turbines. Using locations of GPS-tagged bearded vultures, a rare scavenging raptor reintroduced into the Alps, we built a spatially explicit model to predict potential areas of conflict with future wind turbine deployments in the Swiss Alps. We modelled the probability of bearded vultures flying within or below the rotor-swept zone of wind turbines as a function of wind and environmental conditions, including food supply. Seventy-four per cent of the GPS positions were collected below 200 m above ground level, i.e. where collisions could occur if wind turbines were present. Flight activity at potential risk of collision is concentrated on south-exposed mountainsides, especially in areas where ibex carcasses have a high occurrence probability, with critical areas covering vast expanses throughout the Swiss Alps. Our model provides a spatially explicit decision tool that will guide authorities and energy companies for planning the deployment of wind farms in a proactive manner to reduce risk to emblematic Alpine wildlife.

**Author for correspondence:**
Sergio Vignali
e-mail: sergio.vignali@iee.unibe.ch

# 1. Introduction

The evident negative impacts of global warming on our economy and the biosphere have led many countries to tackle the energy crisis and define objectives for reducing their fossil fuel consumption. Recently, the European Union has launched the 'European Green Deal' [1], a new broad strategy that aims to drastically reduce greenhouse gas emissions and decouple the economic growth from carbon emissions. The goal of this EU initiative is to become 'climatically neutral' by 2050 and this will involve, among others, the extensive use of renewable sources of energy. Yet, along with the claim to progressively rely exclusively on clean energy, another main target of this new deal is to preserve biodiversity. Wind energy, together with solar energy, is likely to lead the green energy revolution worldwide [2]. However, its negative effects on biodiversity have been the subject of an intense debate [3–7], known as the 'green–green dilemma' [8–11]. Indeed, even though wind energy can help to reduce greenhouse gas emissions, it may represent a new threat to sensitive wildlife whose protection may in turn hamper the development of this energy sector.

Some of the species most affected by the large-scale expansion of the wind industry are already of conservation concern, particularly flying vertebrates such as birds [6,12] and bats [13,14]. Large soaring diurnal raptors are often the main avian victims of the blades of wind turbines [15–19]. This is particularly concerning as they have a late sexual maturity and a low reproductive rate. Hence, even a slight increase in their mortality rates can exert strong negative impacts on their population dynamics [20–23]. Wind energy facilities are often erected in regions where landforms and climate generate favourable conditions to support the soaring flight of vultures [24,25], either via thermal or orographic updraughts. The limited frontward visual field of diurnal raptors [26], which reduces their ability to perceive obstacles appearing in their direction of movement, especially when foraging, further exacerbates collision risks. Moreover, when rising in a wind updraught, a raptor follows an ascending spiral (in case of thermals) or a figure-of-eight-shaped path (in case of orographic updraughts) [27], which may suddenly expose it to a rotating blade that was still invisible a few seconds ago due to the frequent change of flight azimuth [28].

Reconciling the transition towards a genuinely greener energy production thus necessitates rigorous and strategic planning that satisfies the dual objective of executing this transition without jeopardizing wildlife survival. To prevent detrimental impacts of the turbine operation on endangered species, wildlife managers and wind energy companies need adequate planning tools to minimize the deployment of wind facilities in areas where major conflicts with biodiversity preservation will occur. Different approaches have been used as planning tools to mitigate the risks encountered by flying vertebrates, spanning from mere delineations of buffer areas around sensitive locations [29–31], through the compilation of distribution areas of sensitive species [28,29,32], to more complex methods that account for fine-grained habitat use and/or flight behaviour of potentially impacted bat and bird species [13,33–37]. The first approach is fairly imprecise. For example, buffer areas are often created around nesting locations while neglecting habitat selection at other life stages. It is furthermore static, being incapable of accounting for range expansion caused by increases in population sizes of potentially affected species [38]. The second method equates species presence with areas of potential conflict. This approach remains coarse as it does not account for actual fine-grained species–habitat associations, yet it can be valuable for identifying broad areas of potential conflicts. The third method is the most sophisticated and also the most informative. Spatially explicit predictive models allow extrapolation to areas for which data about species presence may be deficient. Moreover, when relying on individual-based data such as GPS-tracking, it enables delineating areas of potential conflict with an unprecedented precision, most notably when providing information about the altitude above ground at which birds fly. This approach opens the door towards three-dimensional spatial modelling aimed to mitigate if not avoid conflicts between flying vertebrates and future wind facilities development.

The aim of this study was to predict areas of the Swiss Alps where bearded vultures (*Gypaetus barbatus*) are likely to fly within or below the critical rotor-swept zone of modern horizontal axis wind turbines. The bearded vulture is a long-lived scavenger listed as vulnerable in Europe [39] and critically endangered in Switzerland [40]. Extirpated from many European countries in the early twentieth century [41], the species has been reintroduced into the Alps since the 1980s, with a steadily growing population that progressively recolonizes its former historical range [42]. Several cases of collisions (including fatalities) with anthropogenic structures have been reported in this re-established population [43,44]. Such collisions include wind turbines, which may represent a growing source of hazard into the future [45]. This is concerning, because even a slight increase in mortality may push

the Alpine population of bearded vultures below demographic self-sustainability [23]. Previously, we developed a spatial model to predict the potential distribution of the species, including its future expansion across the Swiss Alps [45]. We now expand that model to include a vertical dimension, specifically flight altitude with respect to the rotor-sweep zone. This refines our previous modelling projections by incorporating the actual use of airspace.

Using a large dataset of GPS locations collected from tagged individuals, we (i) modelled the probability that bearded vultures fly within and below the critical rotor-swept zone of wind turbines and (ii) identified the environmental and topographic variables that drive flight altitude selection. The model was projected to the entire Swiss Alpine range and combined with the previously modelled potential distribution of the species [45] in order to show the joint probability of bearded vultures flying at risky altitudes within suitable habitat. The resulting map provides useful spatial information to delineate areas where the species would be at risk of colliding with wind turbine blades and therefore represents a useful decision tool for planning the deployment of wind power plants across the Swiss Alpine range while minimizing their potential impacts on emblematic biodiversity.

# 2. Methods

## 2.1. Study area and environmental variables

We modelled the flight altitude of bearded vultures using environmental variables that represent land cover characteristics, geology, topography, food availability and wind conditions (table 1). As a result of a large-scale reintroduction programme, the species is distributed throughout the Alps and regularly breeds in the French, Swiss, Italian and Austrian sectors. We restricted our analysis to the Swiss Alpine range, defined as four of the six biogeographic regions of Switzerland [49]: Northern Alps, Inner Western Alps, Inner Eastern Alps and Southern Alps, since Switzerland hosts most of the breeding territories compared with the other countries [50] and is representative of the different habitat types available in the Alps. We focused on Switzerland as the environmental information required to model the flight altitude differed between all four countries in terms of availability, quality and resolution, and trying to homogenize the different datasets would have introduced biases.

Land cover information was extracted from the digital cartographic model of Switzerland (Vector25, https://www.swisstopo.admin.ch/en/geodata/maps/smv/smv25.html). This vector layer was converted into a raster map with 25 m spatial resolution and reclassified to represent the following 10 classes: orchards, forest, bush, scree, anthropic areas, marshland, water, rock, glacier and remaining areas not included in the other classes (electronic supplementary material, table S1). The geological features were derived from the simplified geotechnical map of Switzerland, which was provided as a digitized vector map by the University of Bern (https://biblio.unibe.ch/maps/bis/publications/dl-oef21.html). Specifically, it represents the types of the topmost rock strata (https://data.geo.admin.ch/ch.swisstopo.geologie-geotechnik-gk200/ [46]). The shapefile was converted into a raster map with 25 m spatial resolution and reclassified into four classes: areas dominated by limestone, granite, gneiss and remaining geological substrates (electronic supplementary material, table S2). Topography was characterized with five raster layers extracted from a digital elevation model with a spatial resolution of 25 m (DHM25, https://www.swisstopo.admin.ch/en/geodata/height/dhm25.html). The aspect of the study area was represented by the deviation from east and north (*sine* and *cosine* of aspect, respectively). Terrain characteristics were incorporated by using the topographic position index (TPI, Wilson [47]) and the slope unevenness, which describe the elevation or slope of a cell relative to the surrounding terrain, respectively (both calculated within a moving window of nine pixels). Northness and eastness were calculated with ArcGIS 10.2, and TPI and slope unevenness were derived using the *raster* package in R [51]. Food availability was described using the modelled probability of chamois and ibex occurrence, the two main sources of food for bearded vultures, which thus served as a proxy for food supply (for methodological details see [45], electronic supplementary material, Appendix A). Finally, average wind speed at 100 m.a.g.l. with a spatial resolution of 100 m was extracted from the Swiss Wind Atlas [48]. Pairwise Spearman's correlations between all continuous environmental variables were $|r_s| < 0.6$, calculated based on 10 000 random locations. Categorical variables (i.e. land cover and geology) were one-hot encoded while continuous variables were normalized using the mean and standard deviation derived from the training dataset.

**Table 1.** Environmental predictors used to model the probability of bearded vultures flying below 200 m.a.g.l. (i.e. within the flight altitude range swept by wind turbine blades) across the Swiss Alps, with indication of unit of measurement, abbreviation and data source.

| category | description | unit | abbreviation | source |
|---|---|---|---|---|
| land cover | | | landcover | Vector 25[a] |
| geology | | | geology | gk200[b] |
| topography | *sine* of the aspect | −1 to 1 | eastness | DHM25[c] |
| | *cosine* of the aspect | −1 to 1 | northness | DHM25 |
| | slope | degree | slope | DHM25 |
| | slope unevenness | index | slope_unev | DHM25 |
| | topographic position index[d] | index | tpi | DHM25 |
| food | ibex occurrence probability | 0–1 | ibex | Vignali *et al.* [45] |
| | chamois occurrence probability | 0–1 | chamois | Vignali *et al.* [45] |
| climate | average wind speed at 100 m above ground | m s$^{-1}$ | windspeed | BFE[e] |

[a]Digital cartographic model of Switzerland: https://www.swisstopo.admin.ch/en/geodata/maps/smv/smv25.html.
[b]Simplified geotechnical map of Switzerland [46].
[c]Digital height model of Switzerland: https://www.swisstopo.admin.ch/en/geodata/height/dhm25.html.
[d]Topographic position index according to Wilson [47].
[e]Swiss Wind Atlas [48].

## 2.2. Species data and data processing

Between 2005 and 2020, as part of the Alpine reintroduction programme, 97 bearded vultures were equipped with GPS loggers (battery or solar-powered). GPS devices were fitted with a leg loop harness [52] and birds were released at several release sites in four different countries: Austria, France, Switzerland and Italy. All birds but one were tagged as fledglings; 81 were captive-bred and 16 wild-hatched. In addition, one adult bird, released in 1999, was tagged in 2017 after recapture, rehabilitation and re-release. Loggers from different manufacturers and relying on various power sources were deployed, and GPS locations were collected with a very heterogenous schedule. For example, some devices were programmed to collect bursts with high frequency resolution (1 Hz) as long as the bird was moving and the battery was sufficiently charged. Others collected GPS locations at 1 min resolution under similar conditions, while some devices recorded data with even lower temporal resolution. Since we were interested in modelling the flight altitude above ground level, we selected only data collected by GPS devices that simultaneously recorded information on both flight altitude and instantaneous ground speed so that non-flight locations could be excluded from the analysis (see below). Some of the devices provided flight altitude estimates relative to the mean sea level while others measured it relative to the earth ellipsoid. In the latter case, flight altitude measures were converted to altitude relative to the mean sea level using the method described by Poessel *et al.* [53] (electronic supplementary material, Appendix S1). The altitude relative to the mean sea level was then used to estimate the flight altitude above the ground level by subtracting the ground elevation extracted from the digital elevation model at each GPS location.

Several authors described the problem of negative flight altitude values estimated from GPS locations (see for example [53–56]). Negative flight altitude values are essentially due to the sum of errors in the measure of the altitude and/or position provided by the GPS tag and in the interpolation of the digital elevation model used to calculate the flight altitude relative to ground level. Visual inspection of GPS locations collected at 1 Hz resolution showed that most negative flight altitude values occurred close to steep slopes suggesting a significant influence of the position error in generating negative values. To reduce the position error we culled our data by removing observations with a horizontal dilution of precision (HDOP) ≥ 10 (when the HDOP was provided), which correspond to an error of about 30 m [55] or by discarding all locations with an error greater than or equal to 30 m using the position error provided by the manufacturer. Moreover, we retained only locations with a flight altitude above ground level within the range of −50 to 4000 m [24], assuming that values outside of this range were probably generated by an erroneous measure of the flight altitude.

In order to ensure that the locations retained for our analysis were all collected from flying vultures we considered a combination of two criteria. First, we selected only GPS locations recorded during the day, from sunrise to sunset, using the R package *suncalc* [57]. Second, we removed all locations whose instantaneous ground speed was less than $2 \, \text{m s}^{-1}$ and flight altitude was less than 100 m.a.g.l. [24]. Thresholding in this manner might have removed some valid flying positions, but we preferred to be conservative and avoid the risk of including non-flight locations. Finally, for each bird we selected only observations collected within the Swiss Alpine range, removed all GPS locations recorded during the first eight weeks after fledging to reduce a potential bias related to the release event, and sampled one observation per minute in the case of bursts collected at 1 Hz resolution. This last step was necessary to avoid an overrepresentation of the vultures that collected high temporal resolution data. All birds for which at least 100 locations were available after the filtering process have been retained for the analysis.

## 2.3. Modelling approach

Tracking animals with GPS devices has expanded over the last years and new generation loggers are able to collect many locations at fine temporal resolution [58,59]. Processing a large amount of data is challenging and computationally expensive. Often data are heavily subsampled not only to reduce autocorrelation problems but also to meet computational capacities of classical statistical approaches [24,33]. On the other hand, more recent techniques that require large datasets, like machine learning algorithms and, especially, artificial neural networks, can capture complex nonlinear relationships present in the data. In this regard many tools have been developed to speed up computation with graphics processing unit (GPU) acceleration and create data pipelines to efficiently pre-process data before model training. Artificial neural networks gained popularity in many fields of biology during the last decade [60]. For example, they have been used for behavioural classification from tri-axial acceleration data [61–63] or from GPS data [64,65] and to model datasets with high temporal resolution [63,64,66].

In order to make use of all information included in the data and also develop a method that easily scales to potentially very large datasets, we used a deep feedforward neural network to model the probability of a bearded vulture flying within a given altitude range at a given location. Considering the still ongoing trend of increasing heights of newly constructed, modern wind turbines, we decided for a threshold of 200 m (hereafter referred to as critical altitude), below which the flight of a bird is deemed to be at potential risk of collision with the rotor blades (see also [33,34]). The flight altitude was converted to a binary response with 1 being a location within the critical altitude range and 0 otherwise. Our model was defined and trained within the tensorflow framework [67] and using the *keras* R package [68]. The Keras application programming interface (API) allows great flexibility in defining the architecture of a neural network. We used two hidden layers connected by a dropout layer and a single unit as output of the network that used a sigmoid activation function (for model implementation, see R code in Dryad Digital Repository [69]). A dropout layer acts as a regularization layer by randomly deactivating some units during training, thus reducing the risk of overfitting the training data [70]. The model was trained to minimize the binary cross-entropy loss function using the Adam optimizer.

During the modelling process, we first conducted a grid search experiment to identify the best model architecture, varying the number of units in the hidden layers independently from 16 to 512, each time doubling the number of units (i.e. 16, 32, 64, 128, 256 and 512), and searching the rate of the dropout layer in the range 0.2–0.7 with increments of 0.1. This resulted in a total of 216 different model configurations. We trained the model on 70% of the data, used the remaining 30% for validation and stopped model training when the area under the receiver operating characteristic (ROC) curve (AUC) [71] computed for the validation dataset did not increase for more than 10 epochs.

In a second step, we used a cross-validation approach to evaluate the ability of the model to generalize among different individual birds and for different zones of the study area. Using the best model configuration, identified with the random search experiment, we trained 30 different models for 20 epochs, each time leaving out the locations collected from a different bird on which the model predictions were then evaluated. Similarly, we ran a spatial block cross-validation dividing the study area into spatial blocks created with the *blockCV* R package [72]. The analysis of the spatial autocorrelation among continuous variables, conducted using the function *spatialAutoRange*, suggested a block size with a minimum side-length of 5719 m. We used 10 km blocks (electronic supplementary

material, figure S1) to verify the ability of the model to generalize across wider areas. Using the blocks we randomly partitioned the GPS locations within them, into five cross-validation folds.

As a third step, we investigated the contribution and the marginal effect that each environmental variable had on the model predictions. The contribution of different variables was estimated via their permutation importance using the *vip* R package [73] and measuring the drop in AUC, while the marginal effect of the environmental variables was investigated using individual conditional expectation (ICE) [74] and partial dependence (PD) [75] plots created with the *pdp* R package [76]. ICE curves are generated for a given variable in the dataset, the range of which is subdivided into a grid of *n* equally spaced values. For one observation in the dataset, predictions are made by varying the focal variable within the grid while the other variables are kept constant, thus creating a single ICE curve. The process is repeated for each observation, generating as many ICE curves as there are observations in the dataset. This is a good method to show complex interactions among variables while the overall effect is shown by the PD curve, which simply represents the average of all ICE curves.

Finally, we evaluated model uncertainty by means of a bagging procedure [77]. We sampled the training dataset (70% of the locations) 30 times, with replacement, and used the validation dataset to stop model training when the validation AUC did not increase for more than 10 epochs. We then used the 30 trained models to project predictions to the full extent of the study area and used the mean of the 30 produced maps as final prediction. A 95% credible interval was also created to identify areas were model predictions are more uncertain. The full extent prediction was visually evaluated by field experts (R.A., D.H. and F.L.) to verify that known areas where the species flies close or far from the ground were correctly identified by the model.

## 2.4. Conflict map for landscape planning

The map obtained from the above modelling represents the probability of a bearded vulture flying within or below the critical zone swept by turbine blades, regardless of the habitat conditions being suitable for the bearded vulture (i.e. independent of the probability of species occurrence). To identify areas within actual species' suitable habitat in which there would exist a risk of collision in case of wind turbine installation, we combined the output of our previously developed habitat suitability model (see Vignali *et al.* [45], fig. 4e, reported also in figure 3c) with the output of the model described in this article. The joint probability of species occurrence and flying within the critical altitude range (figure 3e) was calculated by taking the product of the two raster maps [34].

We also converted the predicted probabilities of a beaded vulture flying at risky altitudes into a binary map by means of the threshold which held a sensitivity of 95% (electronic supplementary material, figure S2). The resulting binary map was then intersected with the potential conflict map described in Vignali *et al.* [45] (fig. 4d, reported alse in figure 3d) in order to delineate the areas within the habitat where the species flies within the critical altitude range which would be particularly prone to collisions (figure 3f). Hereafter this synthetic map is referred to as high-risk conflict map.

The whole analysis was run in R [78] v. 4.0.2 through the RStudio software [79].

# 3. Results

## 3.1. Tracking data

After filtering, we retained data from 28 bearded vultures tagged in Switzerland, France and Austria and tracked from September 2014 to December 2020. The quantity of collected locations, as well as the number of tracking days, varied significantly among tagged individuals, with larger sample sizes in birds released within Switzerland, and lower sample sizes in birds that only occasionally visited the study area, stemming from release sites in the neighbouring countries (table 2). The number of tracking days within the Swiss Alpine range varied from 5 to 1411 per individual while the duration of the tracking period per individual varied according to the lifetime of the solar-battery system, any device loss or deficiency, or in the case of a bird's death. A total of 3 040 584 GPS locations were retained after data cleaning, of which 73.9% were collected below 200 m.a.g.l. (average proportions varying between individuals from 57.5% to 89.0%, table 2). After subsampling the GPS locations collected at 1 Hz resolution, flight altitude was finally modelled based on 221 209 GPS locations.

**Table 2.** GPS-tagged birds included for modelling the flight altitude of bearded vultures in the Swiss Alps with country of first release (or subsequent recapture), origin (C: captive-bred; W: wild-fledged), year of fledging, sex (M: male; F: female; U: unknown), manufacturer of the transmitter, number of GPS locations retained after data cleaning ($N$), date of the first (start) and last (end) GPS fix of the cleaned data, total number of tracking days within the Swiss Alpine range, per cent of locations below 200 m.a.g.l. (%), number of GPS locations retained after subsampling one location per minute ($S$) and inter-fix interval (in seconds) given as median and minimum maximum range.

| bird ID | country | origin | year | sex | manufacturer | N | start | end | days | % | S | inter-fix interval |
|---|---|---|---|---|---|---|---|---|---|---|---|---|
| BG1071 | CH | C | 2020 | F | e-obs | 5324 | 8 Sep 2020 | 31 Dec 2020 | 103 | 79.9 | 1727 | 600 (60–23 982) |
| BG1068 | CH | C | 2020 | M | e-obs | 18 611 | 6 Sep 2020 | 29 Dec 2020 | 90 | 89.0 | 1979 | 595 (60–27 618) |
| BG1003 | CH | C | 2018 | F | e-obs | 850 243 | 19 Sep 2018 | 31 Dec 2020 | 675 | 83.2 | 24 711 | 60 (60–28 199) |
| BG1001 | CH | C | 2018 | M | e-obs | 551 350 | 1 Sep 2018 | 17 Dec 2020 | 607 | 76.4 | 18 864 | 300 (60–28 186) |
| BG964 | CH | C | 2017 | M | e-obs | 978 853 | 19 Aug 2017 | 31 May 2020 | 706 | 74.4 | 27 618 | 60 (60–37 213) |
| BG899 | CH | C | 2016 | M | Microwave | 43 050 | 5 Aug 2016 | 20 Nov 2020 | 631 | 74.8 | 33 287 | 71 (60–29 170) |
| BG900 | CH | C | 2016 | M | Microwave | 17 994 | 9 Aug 2016 | 17 Jan 2018 | 307 | 72.4 | 12 802 | 71 (60–31 914) |
| BG841 | CH | C | 2015 | F | Microwave | 7565 | 12 Aug 2015 | 23 July 2020 | 411 | 69.1 | 6061 | 121 (60–31 059) |
| BG838 | CH | C | 2015 | F | e-obs | 1490 | 14 Aug 2015 | 23 March 2020 | 361 | 77.3 | 1461 | 915 (283–28 800) |
| BG802 | CH | C | 2014 | M | Microwave | 41 883 | 3 Sep 2014 | 10 May 2020 | 1411 | 72.2 | 33 205 | 81 (60–46 063) |
| BG797 | CH | C | 2014 | M | Microwave | 13 328 | 10 Apr 2015 | 31 Dec 2020 | 1046 | 72.6 | 11 600 | 143 (60–41 220) |
| BG321[a] | CH | C | 1999 | F | Ornitela | 4230 | 16 Aug 2017 | 16 Dec 2020 | 290 | 57.5 | 844 | 3610 (60–32 349) |
| BG1031 | FR | C | 2019 | F | Ornitela | 3585 | 8 July 2020 | 7 Sep 2020 | 32 | 58.2 | 618 | 900 (60–21 530) |
| BG980 | FR | C | 2018 | M | Ornitela | 11 048 | 11 May 2019 | 19 June 2019 | 27 | 59.7 | 506 | 1200 (60–15 657) |
| BG983 | FR | C | 2018 | M | Ornitela | 6392 | 28 Feb 2019 | 28 June 2019 | 13 | 71.1 | 195 | 664 (60–21 606) |
| BG905 | FR | C | 2016 | M | e-obs | 4014 | 30 Mar 2017 | 3 Apr 2017 | 5 | 73.3 | 265 | 120 (60–3246) |
| W361 | FR | W | 2020 | U | Ornitela | 4834 | 13 Sep 2020 | 22 Nov 2020 | 19 | 68.0 | 254 | 408 (60–21 466) |
| W356 | FR | W | 2020 | U | Ornitela | 219 | 11 Nov 2020 | 31 Dec 2020 | 45 | 75.8 | 219 | 1818 (200–14 426) |
| W346 | FR | W | 2020 | U | Ornitela | 20 551 | 12 Sep 2020 | 20 Nov 2020 | 15 | 81.6 | 575 | 159 (60–12 258) |
| W285 | FR | W | 2019 | F | Ornitela | 141 104 | 11 Sep 2019 | 31 Dec 2020 | 178 | 73.9 | 4659 | 416 (60–28 804) |
| W284 | FR | W | 2019 | F | Ornitela | 30 086 | 30 Jan 2020 | 8 Apr 2020 | 59 | 60.1 | 1379 | 612 (60–21 652) |
| W313 | FR | W | 2019 | F | Ornitela | 11 628 | 18 Apr 2020 | 14 Nov 2020 | 10 | 68.2 | 304 | 171 (60–6672) |
| W251 | FR | W | 2018 | M | Ornitela | 16 955 | 3 Mar 2019 | 23 Dec 2020 | 126 | 59.7 | 956 | 1217 (60–36 020) |
| W209 | FR | W | 2017 | M | Ornitela | 108 854 | 7 Aug 2017 | 9 Oct 2020 | 138 | 68.1 | 3484 | 657 (60–35 958) |
| W196 | FR | W | 2016 | F | Ornitela | 66 985 | 10 May 2017 | 26 Dec 2020 | 1084 | 67.9 | 21 678 | 620 (60–42 894) |
| BG998 | AT | C | 2018 | M | Ornitela | 44 486 | 6 Oct 2018 | 25 Dec 2020 | 291 | 76.6 | 3597 | 930 (60–32 399) |
| BG843 | AT | C | 2015 | M | e-obs | 32 556 | 30 Aug 2015 | 18 May 2020 | 321 | 58.3 | 7920 | 301 (60–28 827) |
| BG840 | AT | C | 2015 | M | e-obs | 479 | 28 June 2016 | 13 March 2017 | 77 | 72.7 | 441 | 1930 (64–18 047) |

[a]tagged as adult bird in 2017.

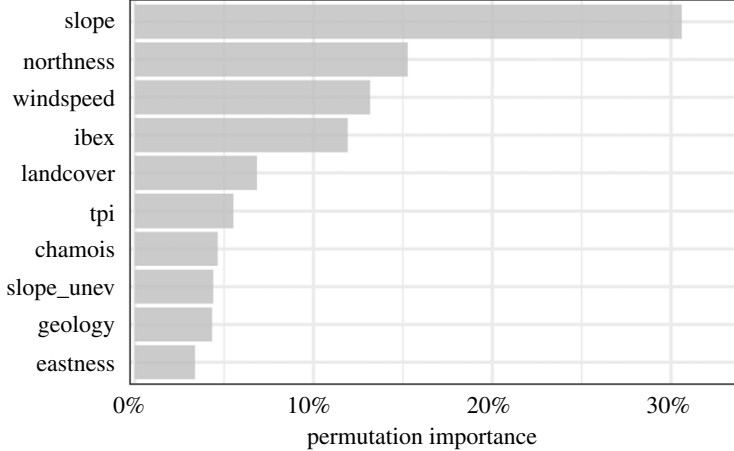

**Figure 1.** Permutation importance of the environmental variables used to model the probability of bearded vultures flying below 200 m.a.g.l. Permutation importance is presented as the drop in training AUC (%) when randomly permuting the values of the respective variable within their empirical range. Variable abbreviations are given in table 1.

## 3.2. Model architecture

The best model configuration identified during the grid search experiment was a deep feedforward neural network with 256 units in the first hidden layer, a dropout rate of 40%, and 32 units in the second hidden layer. This model had an AUC value of 0.730 for the training dataset and 0.708 for the validation dataset. Overall, the model was able to generalize well across birds, which was indicated by a mean training and testing AUC of 0.718 (s.d. = 0.004) and 0.703 (s.d. = 0.037), respectively (electronic supplementary material, table S3), which was comparable to the performance of the model trained using all birds. Similarly, the model showed a good ability to generalize across different regions of the study area with a mean training and testing AUC of 0.713 (s.d. = 0.007) and 0.699 (s.d. = 0.015), respectively (electronic supplementary material, table S4).

## 3.3. Relative contribution of different variables and model predictions

The environmental conditions that mainly drove the probability of a bearded vulture flying within the critical altitude range were steepness of the terrain, aspect, wind speed and food availability (permutation importance of 30.6, 15.3, 13.2 and 11.9%, respectively) (figure 1). Bearded vultures were more likely to fly at lower altitude (less than 200 m.a.g.l.) when approaching steeper slopes of south-facing mountainsides compared with north-facing mountainsides (figure 2). Flying within the critical altitude range was also more likely to occur in areas typically exposed to stronger winds compared with areas with weaker winds. Over areas with a high probability of ibex presence, (i.e. sectors where it is more likely to find ibex carcasses), the probability of flying below 200 m.a.g.l. was always high. This pattern is evidenced not only by the PD curve, but also by the increasing concentration of the ICE curves with increasing values of this variable. Overall, the probability of flying within the critical altitude range was higher over areas dominated by scree and rocks compared with the remaining land cover conditions (i.e. forest, anthropic areas, water bodies, etc.).

The combination of the potential conflict map (figure 3*d*) with binary representation of the probability of flying below the critical altitude of 200 m.a.g.l. (Figure 3*b*) revealed that about 77% of the area suitable for the species is likely to be overflown within the critical altitude range (figure 3*f*). This area, ranging from 267 to 4502 m.a.s.l., represents 30.6% (7878 km$^2$, table 3) of the overall extension of the Swiss Alpine massif.

## 4. Discussion

The wildlife versus wind energy conflict model developed here extends commonly applied approaches of predicting areas of potential collision risk with wind turbine blades based on mere species' spatial occurrences by adding the vertical dimension of flight behaviour. In effect, its predictions are refined by quantifying the areas within which bearded vultures would effectively fly within or below the

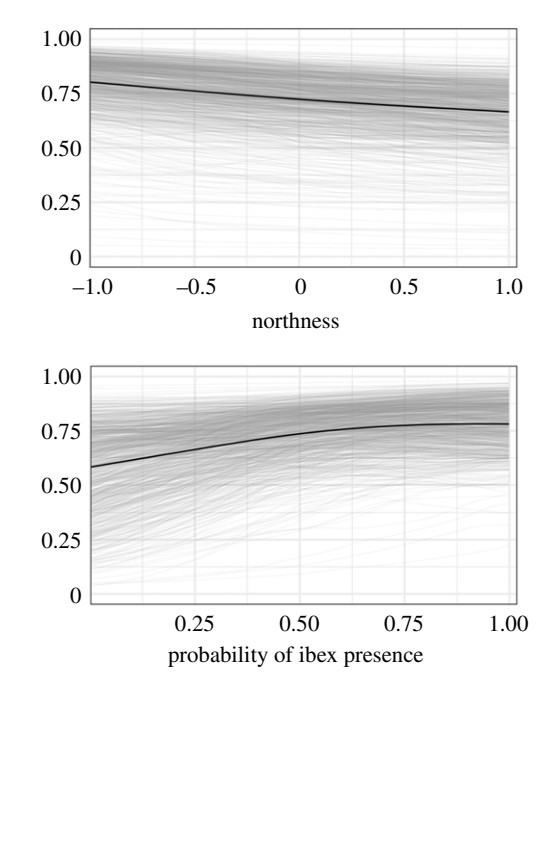

**Figure 2.** Marginal effect of the five most important environmental variables for predicting the probability of a bearded vulture flying below 200 m.a.g.l. In grey are plotted 1000 randomly sampled individual conditional expectation (ICE) curves [74] and in black the partial dependence (PD) curve [75]. For land cover, a categorical variable, each boxplot shows the ICE values without outliers and the black dot the value of the PD. The curves for the remaining environmental variables are shown in electronic supplementary material, figure S4.

vertical segment of airspace swept by the blades of potential wind turbines. The projected map showing the probability of bearded vultures flying below the critical altitude (figure 3a) revealed that wide areas of the Swiss Alps are potentially prone to collisions. The probability of flight at risky altitudes was particularly high along mountainsides and ridges. This is not surprising, since valleys are overflown at high elevation, notably during commuting relocations. While this probability map gives a general overview of the areas with environmental conditions favouring low flight altitudes throughout the Swiss Alps, only the map resulting from the joint probability of species occurrence [38] and of flying below the critical altitude range (figure 3e) encompasses the whole complexity of the species–habitat associations, including ecological requirements and flight behavioural routines. By intersecting these two probabilistic maps, we could filter out areas within the species' habitat extension where it is unlikely that bearded vultures would fly within the critical altitude range, (i.e. mainly the valley bottoms), while we managed to highlight critical hotspots of potential conflict with wind energy development.

Steep south-facing slopes dominated by strong winds and areas with high probability of ibex presence (i.e. providing carcasses potentially exploitable by bearded vultures) offered the best conditions for low-altitude flight (figures 1 and 2). Terrain steepness and exposure are indeed two key factors explaining the formation of updraughts, specifically thermals and orographic updraughts. Thermals are generated by unequal heating of the earth surface: solar radiation heats up certain land cover types faster than others (e.g. dark rocky outcrops), thus generating columns of warm air that

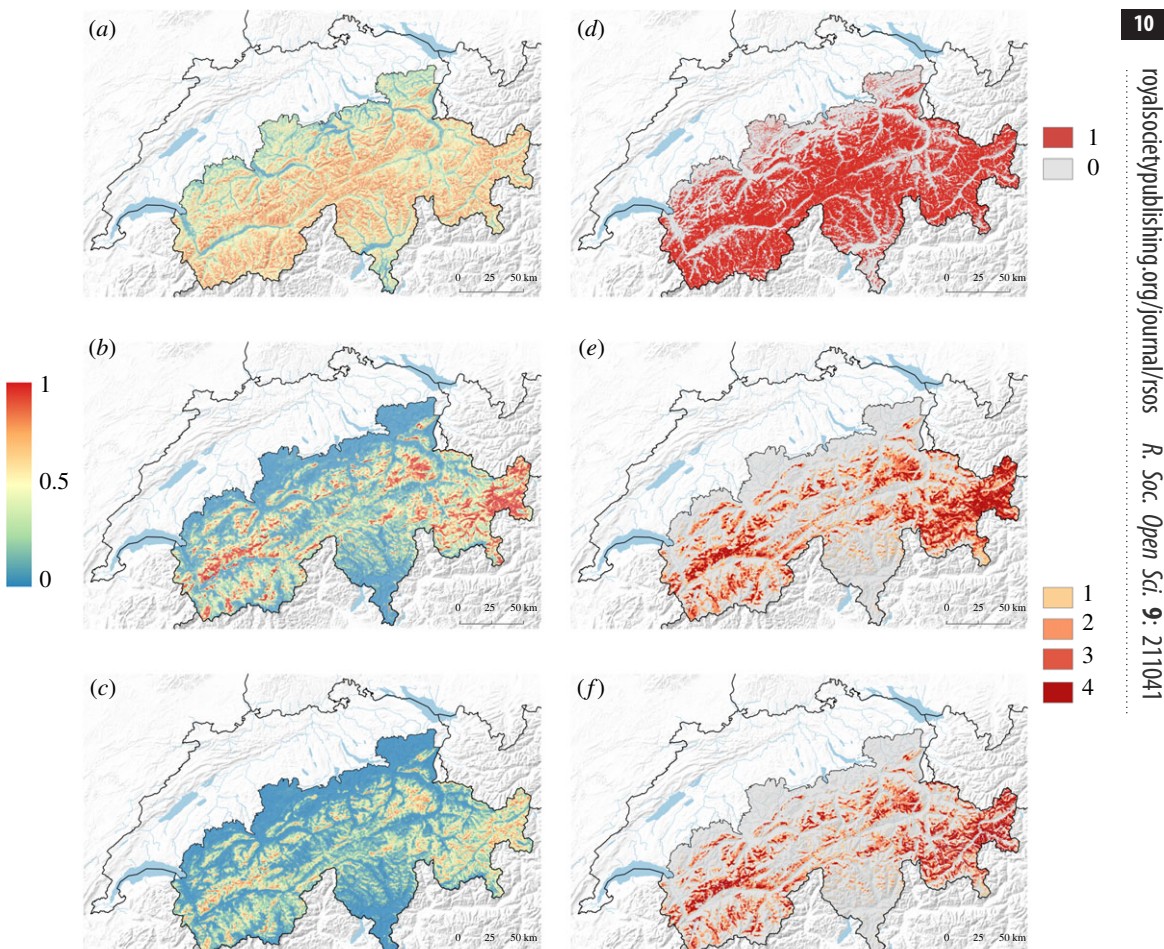

**Figure 3.** Graphical representation of the research approach used to model risk to bearded vultures from wind turbines. Maps show the data layers combined to produce the risk maps that are the final product of that modelling exercise. The maps show (*a*) the predicted probability of a bearded vulture flying below 200 m.a.g.l. calculated as the mean prediction of a 30-bagging procedure and extrapolated to the whole Swiss Alpine range; (*b*) the probability of bearded vulture occurrence described in fig. 4*e* in [45]; (*c*) joint probability of occurrence and flying below 200 m.a.g.l., calculated as the product of maps (*a*) and (*b*). These probability maps are shown with a gradient ranging from blue: zero probability, to red: high probability. The map shown in (*d*) is the translation of map (*a*) into a binary response using the threshold for which 95% of the locations occurring at risky altitudes are correctly predicted (the areas with a high probability that a bearded vulture flies within the critical altitude range are shown in red); (*e*) the 'potential conflict map' described in fig. 4*f* in [45] with increasing risk represented by an increasing intensity of red (see [45] for further explanation); (*f*) the 'high-risk conflict map' calculated as the product of (*d*) and (*e*). The Swiss Alpine range is represented in light grey in (*d*), (*e*) and (*f*).

rise from the ground. Orographic updraughts stem from the upward deviation of wind blowing against mountainsides. Although the former occur mostly along sun-exposed rocky slopes and over flat areas [80], the latter are generated exclusively along slopes and ridges. While thermals in lowlands are furthermore easily disrupted by dominant winds, mountain regions supply a year-round source of updraughts, thus providing optimal conditions to support the flight of raptors exhibiting high wing-loading such as vultures [81]. In addition, orographic updraught tends to subside with altitude, thus keeping soaring birds flying at lower altitudes than when exploiting thermals [55]. In line with this, various studies have shown that raptors fly relatively low over ridges and steep slopes [33,55,82] where orographic updraught is more likely to occur. As shown here, bearded vultures are no exception to that rule.

While static and/or dynamic environmental conditions are frequently included to model flight behaviour in relation to potential collision risk with wind turbines [33,34,54,55,82], the spatial distribution of natural food supply has—to the best of our knowledge—never been considered so far (but see [34] which considered the distance from artificial feeding sites). Our model ranked this environmental variable with an importance similar to aspect or wind conditions in explaining the

**Table 3.** Percentage of shares of the different levels of sensitivity (increasing from 1 to 4) predicted in the potential conflict map (figure 3e) and the high-risk conflict map (i.e. where the bearded vulture is likely to fly within the critical altitude range, i.e. below 200 m.a.g.l., figure 3f) in the whole Swiss Alps. The last column reports the shares of areas where the bearded vulture is likely to fly above the critical altitude within the habitat.

| level of sensitivity | potential conflict map | high-risk conflict map | remaining |
|---|---|---|---|
| 1 | 12.3 | 8.4 | 3.9 |
| 2 | 11.0 | 8.5 | 2.6 |
| 3 | 7.2 | 5.8 | 1.4 |
| 4 | 9.2 | 7.9 | 1.2 |
| total | 39.7 | 30.6 | 9.1 |

probability of flying within the critical altitude range. Vultures, and large soaring birds in general, are constrained in their movements by the availability of updraughts. By soaring into ascending air currents for gaining altitude and then gliding to another location [83], they achieve an energy-efficient commuting pathway [84,85]. However, the optimal altitude to be gained with soaring is likely to result from a trade-off between mobility for long-range horizontal displacement and the ability to inspect the ground for locating food, the latter diminishing with altitude [86]. This might be particularly crucial for bearded vultures which, given their peculiar bone-based diet [87,88], have to locate carrion and small parts of carcasses that may be easily overlooked. Ibex carcasses represent the most important food supply of Alpine bearded vultures, to an extent that their distribution is largely driven by the presence of that ungulate species [42,45]. Our model emphasizes the importance of that ecological requirement, since the probability of flying within the dangerous altitude range was always high over areas with high probability of ibex presence, regardless of the other environmental conditions (figure 2). That a similar pattern was not found for chamois, another important food source for Alpine bearded vultures, is not surprising, because chamois' distribution is much more uniform, as the species is less dependent on a rocky substrate than ibex. Several studies have shown that food resources affect habitat use of vulture species [34,89–93]. Here, we have demonstrated that the spatial distribution of the main source of food can also affect the flight altitude behaviour. Investigating the effect of food supply on flight altitude behaviour may, therefore, help to improve the estimate of potential conflicts with wind energy development, also for other raptor species that are more broadly distributed across the landscape.

Given that the majority of the GPS locations (73.9%) indicated flight activity at altitudes lower than 200 m.a.g.l. (electronic supplementary material, figure S3), bearded vultures in the Swiss Alps seem to be active most of the time in the dangerous altitude range. This concurs with former findings by Rushworth & Krüger [25] who estimated that south African bearded vultures spend 74.6% of their foraging time below 200 m.a.g.l. Similarly, Reid *et al.* [34] found that bearded vultures in South Africa spent the majority of their time below the critical altitude (66% and 55% for non-adults and adults, respectively). As a result, our final high-risk conflict map shows that 77.0% of the area of suitable habitat across the Swiss Alps (i.e. 30.6% of the entire Swiss Alpine range) may incur some potential collision risk (figure 3f, table 3).

We used two different validation approaches to assess the ability of our model to generalize across individuals and geographical regions within the Swiss Alps. In this respect, the leave-one-bird-out cross-validation is important to ensure that model predictions are not biased by individuals with a prevailing number of GPS bearings. Only five birds out of 28 (i.e. W361, BG980, BG998, BG797 and BG843, electronic supplementary material, table S3) caused some marked drop of the AUC when used to evaluate the model. These notwithstanding, a few birds yielded GPS locations exclusively from one part of the study area (e.g. only in western, central or eastern Swiss Alps), while most of them roamed across the entire Swiss Alpine massif. The similarly good AUC values obtained across the individuals confirm the reliability of our model extrapolations to the entire study area. Finally, the spatial block cross-validation further reinforced the validity of the model throughout the study area, given that it performed equally well for predicting the locations in the spatial blocks not used for model training [94].

The AUC values yielded by validation were not particularly high, although comparable with those obtained by Reid *et al.* [34] for the bearded vulture population inhabiting southern Africa. A model

with an AUC greater than 0.7 is considered to be sufficiently accurate for discriminating positive from negative classes (i.e. distinguishing between locations where bearded vultures flew below versus above the critical altitude range) [95]. Since bearded vultures may fly both above and below the critical altitude range within any given place, there will always be situations where the model correctly predicts one class and therefore incorrectly predicts the other, resulting in low AUC values. In addition, there are plenty of situations for a mountain raptor, especially near sheer cliffs and steep slopes, where a small horizontal displacement can significantly change the altitude above ground level, and thus the respective flight altitude class. Similarly, even tiny inaccuracies in GPS bearings collected in the vicinity of cliffs can affect the allocation to one of the two altitude flight classes. The AUC is a valid threshold-independent metric useful to evaluate the overall model performance—which is the reason why we relied on it to tune the model's hyperparameters and to check the ability of the model to generalize across birds and areas. However, one always benefits from a final, complementary validation by visually inspecting how the model performs in known areas where the birds have been regularly observed. Finally, a bagging procedure was carried out, which showed that model predictions remained stable over repetitions (electronic supplementary material, figure S5).

Selecting a threshold to convert a probability map into a binary map is always a critical step because it eventually determines the classification skills of the model. The Youden index, maximizing sensitivity plus specificity, is probably the most frequently used threshold approach for classifying species distribution models [96]. It has already been applied to flight altitude models [33]. However, when the conservation of an endangered and vulnerable species is at stake, the ability of a model to correctly predict the risk-class is the most important aspect, even if it comes at the expense of the accuracy in predicting the other class. Applying the principle of precaution, we, therefore, chose a threshold that held a true positive rate of 95% since we considered it especially important to correctly predict locations where the species flew within the critical altitude range. By doing so we were fairly conservative, accepting the risk that some flight locations above 200 m.a.g.l. were wrongly classified into the critical flight altitude range. The resulting probability map (figure 3b) might thus overrepresent critical areas, which is a minor issue from a conservation and risk assessment viewpoint. Therefore, the map showing areas with high risks of conflict (figure 3f) is a refinement of our previous conflict map [45] since it subtracts the areas over which the species is likely to fly high above the ground. Although we adopted a conservative approach, our results suggest that bearded vultures fly above the critical altitude range in about 9.1% (2351 km$^2$) of the areas previously classified as having high conflict potential.

A potential limitation of our study, however, may be the under-representation of adult birds in the dataset. Bearded vultures reach adult plumage at the age of 5–6 years [97]. As they develop from the juvenile (less than 2 years) to the adult age, some morphological characteristics that may influence flight behaviour gradually change [98]. Our dataset includes four birds that collected data up to 6 years and two birds that reached the adult age (6 years according to [98]). Moreover, 10 birds (i.e. more than a third) recorded data for more than 3 years, the age at which immature bearded vultures change their flight behaviour, shifting from an exploring phase to a phase of territory establishment [42,45]. Despite adults being less represented than the other age classes, there are no prominent morphological differences between subadult and adult birds that would suggest differences in their flight altitude patterns. In line with this, other authors found that in soaring raptors the bird's age influences its ranging behaviour [34,99] but not its flight behaviour [100,101].

Our model was developed in a framework particularly suitable for very large datasets. In effect, modern tracking devices are capable of collecting data at high temporal resolution, thereby introducing new challenges for their analysis [58,59]. The use of artificial neural network approaches is a possible solution to address this challenge. First, contrary to other classical statistical methods, it does not require an *a priori* definition of the functional forms for each relationship between predictors, [102] as complex nonlinear relationships among variables are learned directly from the data. Second, it can take advantage of specific libraries developed to create efficient data pipelines (see for example the *tensorflow dataset* library and its R implementation [103]). Data pipelines serve for transformations like normalization, standardization and one-hot encoding of categorical variables on batches of data that are then fed into the neural network. This way it is not required to apply each transformation to the entire dataset but rather to the single batches, and the data can be loaded in batches directly from a file or database (see R code in Dryad Digital Repository [69]).

Although innovative in several respects, our approach focuses only on one species potentially affected by wind turbine deployment. We believe, however, that a similar method could be readily applied to any other raptor species, if not to other soaring birds such as storks or herons, for which flight altitudes might

also be decisive. Of course, a combination of species-specific predictive models obtained from different emblematic species potentially impacted by the wind industry development would be a major step towards a biodiversity-friendly spatial planning. Policy makers and land-use planners, wind energy promoters and conservation biologists would all benefit from the rapid development of such comprehensive decision tools. Wind energy companies, in particular, could evaluate from the onset whether their investments would be at risk of not obtaining official approval. It must be explicitly stressed, however, that models such as the one presented here can inform spatial planning but in no way represent substitutes to *in situ* environmental impact assessments that are prerequisites for any infrastructure project development.

Ethics. All the work has been conducted in accordance with relevant national and international guidelines and conforms to the legal requirements. Birds reintroduced in the Alps hatched within an international captive-breeding programme (EAZA Ex situ programmes) and had been tagged before the release. In the case of wild individuals (nine birds from France and one bird from Italy), captures were carried out in compliance with the Ethical Principles in Animal Research. All animals were not marked for scientific reasons but for the monitoring programme of the ongoing reintroduction of bearded vultures in the Alps. The GPS data were provided to the authors by Asters Conservatoire d'espaces naturels de Haute-Savoie (FR), the Parc national de la Vanoise (FR), the National Park Hohe Tauern (AT), the Parco Nazionale dello Stelvio (IT) and the Swiss Foundation for Bearded Vultures (CH). Data from France, Austria and Italy are purely third-party data. The authors D.H. and F.L. are involved in the monitoring programme of the Swiss reintroduction project. However, according to Swiss law, projects involving wild animals that do not meet the definition of animal experiments in Article 3 letter c of the Animal Protection Act are not subject to any authorization procedure under animal protection law, and measures that serve the protection and management of wild animals do not require an animal experiment permit. Thus, protocols, amendments and other resources have been done according to the national guidelines.

Data accessibility. Data and R code necessary to reproduce the analysis are available via the Dryad Digital Repository https://doi.org/10.5061/dryad.m63xsj43g [69].

Authors' contributions. S.V.: conceptualization, formal analysis, funding acquisition, methodology, writing—original draft; F.L.: data curation, writing—review and editing; D.H.: data curation, writing—review and editing; R.A.: conceptualization, funding acquisition, project administration, resources, supervision, writing—review and editing; V.B.: conceptualization, funding acquisition, methodology, supervision, writing—review and editing.

All authors gave final approval for publication and agreed to be held accountable for the work performed therein.

Competing interests. We declare we have no competing interests.

Funding. This research was supported by the Swiss Federal Office for Energy, the Swiss Federal Office for the Environment and the following foundations: Parrotia Stiftung, Margarethe und Rudolf Gsell-Stiftung, Alfons und Mathilde Suter-Caduff Stiftung, WWF Switzerland, Beat und Dieter Jutzler Stiftung, University of Bern Forschungsstiftung, Stiftung Dreiklang für ökologische Forschung und Bildung, Sophie und Karl Binding Stiftung, Stiftung Temperatio, Ernst Göhner Stiftung, Steffen Gysel Stiftung für Natur und Vogelschutz and Samy Harshallanos.

Acknowledgements. We thank the Swiss Foundation for Bearded Vultures, the Nationalpark Hohe Tauern, the Vulture Conservation Foundation, Vautours en Baronnies, LIFE GypConnect, LIFE GypHelp, ASTERS, LPO Grands Causses, Parc Naturel Régional du Vercors, the Parco Naturale Alpi Marittime and the Parco Nazionale dello Stelvio for sharing the GPS data. We are grateful to Olivier Roth for the English translation of electronic supplementary material, tables S1 and S2, and Dr Ian Ausprey for proofreading the article.

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
