## [Peer Review File · Royal Society Open Science]

Review History

RSOS-211041.R0 (Original submission)

Review form: Reviewer 1

Is the manuscript scientifically sound in its present form?

Yes

Are the interpretations and conclusions justified by the results?

No

Is the language acceptable?

No

Do you have any ethical concerns with this paper?

No

Have you any concerns about statistical analyses in this paper?

Yes

Recommendation?

Accept with minor revision (please list in comments)

Comments to the Author(s)

This paper builds a model from GPS tracking data from bearded vultures to predict the likelihood of a vulture flying within the rotor swept area of a turbine (<200 m agl), based on relevant topographic and environmental variables. Authors use this model to predict the probability of flight <200 m agl across the species range in the Swiss Alps. They then use a previously published predictive map of the probability of bearded vulture occurrence in the same region (Vignali et al 2021) to combine the probability of occurrence and flying in the risky zone to produce a high-risk conflict map to guide sustainable development of wind energy away from areas expected to be highly risky to this species.

I must first note in my review that I am not familiar with the modelling method - deep feedforward neural networks and thus have not commented on the section on the modelling approach. However, the model appear to have been well-cross validated, with both between birds and between region performance being tested. It is hard to tell without being familiar with the deep feedforward neural network procedures, but I would usually expect cross-validation methods to be used to inform model selection in addition to model performance, which has not been done and might be an avenue to explore.

Overall, the paper addresses an important and rapidly growing conservation dilemma and provides maps which will be useful for planning new wind energy developments in locations which pose the least collision risk to this species. This is not the first collision risk map for bearded vultures, which has already been achieved by Reid et al. 2015 [Using spatial analyses of bearded vulture movements in southern Africa to inform wind turbine placement, *Journal of Applied Ecology*]. Although referenced in a couple of places, there has not been much mention of this work, and a discussion on the pros/cons of the two methods or how this work is different and/or progresses conservation further would be merited.

My main concern is that the majority of the tracking data has come from juvenile, immature and subadult birds. According to Kruger et al. (2014) [Differential range use between age classes of southern African bearded vultures, *PlosOne*] bearded vultures only reach adulthood at 6 years old, thus only one individual was tracked as an adult in this study. Therefore, I think the authors can only really achieve a conflict map for the sub-adult sector of the population. Further more specific comments are given below.

Statistical methods

I am unable to comment in depth on the primary methodology which I am unfamiliar with - deep feedforward neural network models. It isn't clear to me if different variable combinations have been tested, or if they have all been assumed to be important predictors of flight height. For example, the permutation importance shows that 'eastness' has an importance of around 3%. Has it been tested if the model would be more parsimonious without this variable? Or is this accounted for in the neural network?

L153. "average wind speed at 100 m a.g.l. was extracted from Swiss Wind Atlas". Is this the average across the year? Derived from monthly, daily or hourly means? Can you give more information on how it was calculated and the spatial resolution of this variable? Do you think the average conditions represent the actual conditions available? E.g. an area might be very windy in one season or time of day, but not windy at all at another time. The 'average' for such an area

would not actually capture either of the conditions available. If this is the case, I would recommend making predictions for actual wind scenarios that occur through the year/day and then averaging the predicted value for each location. This would also require associating each fix with the actual conditions when it was recorded for the analyses stage.

Age classes and data

Previous work by the authors which this work relies on (Vignali et al 2021), i.e. the habitat suitability model, was modelled separately for juveniles and adults, as well as summer and winter. It seems that the majority of the tracking data used in this paper comes from juvenile or subadult vultures, with only one adult being tagged and one additional bird making it to 3.5 years, and two others to 2.7 years (this was according to tracking days in table 2 - however, this is possibly 9 individuals collected data >3 years old, see below, but it is unclear which individuals or how much data). Nevertheless, following Kruger et al. (2014) bearded vultures only reach maturity at 6 years. Thus, the work here really can only assess juvenile/subadult risk and this should be made clear. As a result, I would recommend removing the limited adult tracking data from the analyses. Furthermore, the habitat suitability map used in the analyses should be an aggregation of Fig 4a and Fig 4c (from Vignali et al 2021), i.e. also not including adult data.

L320 - "Some devices (N=9) recorded data for more than 3 years. This is approximately the age at which juvenile bearded vultures change their behaviour, shifting from an exploring phase to a phase of territory establishment [40,43]."

From Table 2, only one individual had more than 1095 tracking days in the analyses. However, I understand this might be number of unique tracking days in the analyses - not days/age since tagging, which would also be useful in Table 2. Kruger et al 2014, used the following age classes: juvenile (post-fledging to 2 years), immature (2-4 years), sub-adult (4-6 years) and adult (.6 years), which is quite a big discrepancy from this paper.

Also, how do you think having different ages, from fledgling to subadult might have affected the results?

L307-310 and Table 2- One individual had only one day of tracking. Is it worth keeping this there? What happened to this individual? Is the tracking data likely to represent 'normal' behavior? Further, I see this individual is one of the ones which had lower AUC cross-validation score (L416).

Other

L361-373. Intrigued as to why these results haven't been compared with Reid et al 2015 - who found a total of 66% on non-adult bearded vulture flight <200 m agl.

L369. Rushworth and Kruger 2014 not in reference list.

L382 "the spatial distribution of potential food supply has - to the best of our knowledge - never been considered so far". Although in a slightly different format, Reid et al 2015 included the variable "distance to vulture restaurant" which is equal to the distribution to reliable food sources for bearded vultures.

Minor

L89 Suggest removing "radio-" since it doesn't have the fine precisions that GPS tracking can achieve; particularly not altitude measurements.

L.155 I am surprised TPI and slope unevenness are not correlated - what was their correlation coefficient?

L201. Since you have a bunch of different tags which might record speed slightly differently, I would suggest using a between peaks minimum as a unique cutoff for each individual - following Schaub, et al 2020 [Collision risk of Montagu's Harriers Circus pygargus with wind turbines derived from high-resolution GPS tracking. Ibis]

L208 - one location per minute seems like autocorrelation may still be an issue?

Very minor. The paper would benefit from English proof reading, some very minor points are given here:

L24 “..with future wind turbine deployments...”. Edit plurals.

L30 Change “expansions” to “expanses”

L56. Edit needed to second half of sentence.

L87-88. “...as spatially-explicit predictive models allow extrapolation to areas...”. Grammar edits.

L92. Change ‘gate’ to ‘door’.

L103. suggest changing “new” to “growing”.

L105-106. Suggest restructuring sentence to avoid “if”. Perhaps “Using a spatial model...”?

L151. Changes ‘providers’ to ‘sources’

L225. Comma ‘,’ after ‘very large datasets’.

L287. ..., this <is> regardless of...

L493. “believe”

Review form: Reviewer 2

Is the manuscript scientifically sound in its present form?

Yes

Are the interpretations and conclusions justified by the results?

Yes

Is the language acceptable?

Yes

Do you have any ethical concerns with this paper?

No

Have you any concerns about statistical analyses in this paper?

Yes

Recommendation?

Major revision is needed (please make suggestions in comments)

Comments to the Author(s)

This is a nice paper on risk mapping and it is an advancement over previous risk mapping. The two prior raptor-wind turbine risk maps based on flight behavior (Reid et al. 2015 (citation 33 in the ms) and Miller et al. 2014 (not cited, but I’ve suggested it in the text) don’t do a great job of combining probability of occurrence with probability of risky flight. This paper explicitly does that and so it is a step forward.

There are a number of changes that I think would make this manuscript better. A few of the most important items here:

- I don’t agree with your use of the term “flight height”. It is vogue to use it, but it is also incorrect. You note Peron’s definition of “flight height” (your line 175). I appreciate that he defined this, and I appreciate that you explain how you chose your terms, but his definition leads to sloppiness. Height describes how tall something is. I’ve corrected it a few times in the pdf, up to you and the editors for a final decision.

- I’m curious as to why you limited your modeling to a political region (Switzerland) rather than a biologically defined region. You have lots of evidence that the birds ignore political

boundaries. For your reports to funders, sure, a political region makes sense. But for a biological study, a biological region makes more sense.

- There needs to be a much more detailed description of your data filtering process and the thresholds you use for inclusion in the analysis. You could do this in the SI (if there are space limitations in the ms) but better in the main ms if you can.

- I am concerned that your data set includes birds with only 1 day of data collection and as few as 77 data points and other birds with as many as 1,300 days and 40,000 data points. Despite this, your modeling approach does not include a random effect for bird, which means that individuals are weighted differently. If you can set a random effect for bird, that would be best. However, since it is probably not trivial to include a random effect in your models, you should probably set some threshold for inclusion and justify it, either with citations or with a simulation exercise.

- I'm concerned about autocorrelation in your dataset. I don't know the modeling techniques you describe well enough to understand if they handle autocorrelation well or not. That said, you randomly selected one obs per minute. That means the data are anywhere from 1s to 120s apart. There is still high autocorrelation in data collected at that time interval. As such, your statistical models need to handle autocorrelation. If you were using a more standard modeling approach, I could include some suggestions – certain non-parametric model types handle correlated data, or imposing a correlation structure on your models works, etc. But I really don't know what to suggest in this case – that one is up to you.

- Several spots in the ms could be substantially shorter.

- The literature cited and citations in text have some issues to address. In most places in the text you use numbers (presumably to match journal format), but in a number of places you use author names. Try to clean that up, as this got confusing in a few spots.

Please see the pdf (Appendix A) for some minor comments and some repetition of things I've written below.

Title – I would change “height” to “altitude”

Introduction – In general, I think the introduction is pretty strong. It would be good to shorten it a bit. The last two paragraphs could be combined and shortened a bit – you don't need to go into the methods too much here, for example. You also have two paragraphs that each only have 2 sentences. Generally it is better to have at least 3 sentences in a paragraph. You can shorten and eliminate some of this text to combine. For example, the last 2 sentences of the first paragraph contain information repeated somewhat in the second sentence of the second paragraph and the sentence that starts at line 69. All that said, the intro is strong and relevant to the topic and mostly well cited.

I've put a number of notes in the pdf. A few to expand on here.

- “lift” is a force generated by the wing (the airfoil). So the words “orographic lift” are nonsensical (again, lots of people use this term, but it is incorrect). Simple fix – “orographic updraft”. Also, “thermal updraft” (not “thermal lift”).

- You suggested that we need planning tools to avoid deploying wind facilities in areas where conflicts with biodiversity will occur. Unfortunately avoiding conflict is a dream we are unlikely to realize (look at projections for expansion of wind in the coming 30 years – there will be lots of conflict). We need to talk about “minimization” of conflict, not avoiding it. Easy but important change to the ms.

Methods – I'm curious as to why you made the decision – to limit the modeling exclusively to Switzerland. I get that the funding is focused on one country. That said, I have a hard time believing that your tagged birds never left the country. Clearly also, the alpine zone extends beyond the borders of this one country. This is mostly splitting hairs and is certainly not a reason

to reject the manuscript. But wouldn't it be more reasonable to model a biologically relevant region, rather than a politically relevant region? Perhaps the "contiguous Alpine areas within 500km of known nests of vultures" or something like that?

My question became more relevant when I realized that many of the birds whose data you used were tagged outside of Switzerland. I think there is good rationale for using a biologically defined study area, rather than an politically defined one.

Can you go into a bit more detail as to what is an "anthropic area" in Switzerland? In the USA, we use the term "urban" for areas in cities. But from my perspective, 90% or more of Switzerland is dominated by humans. As such, specificity seems really important when clarifying your land cover classifications.

Line 170-171 - what is the slowest rate of data collection of any GPS devices?

Rather than simply using flight speed as a threshold to remove "non-flighted" points, you should use both flight speed and altitude. A vulture at high altitudes in a thermal might be moving very quickly upwards but very slowly in a horizontal plane. See both Murgatroyd 2021 and Poessel 2018 for this. Murgatroyd says that Schaub et al. 2020 Ibis also does this.

I'm concerned that some of your birds have <10 days of data collection and that you are modeling birds with so few days of data collection in the same batch as birds with more than 1000 days of data collection. I think it would be good to set a minimum threshold for inclusion in your models. Maybe 15 or 20 days? Likewise, set a threshold for a minimum number of points. Maybe 150 or 200 data points, or at least 1 per day over your minimum time period?

OK - having read the results, you say that you had "requirements for being included in the analysis". I missed those in the methods. Can you clarify in the methods what were the requirements for being included in the analysis? I suspect that will address my concerns above. Likewise, be sure to explain whether your filtering of GPS points was before or after you filtered birds.

I don't love your resampling down to 1m data. The resampling itself is fine. However, the bigger problem is that you don't clearly explain the slowest data rate your transmitters collected. 1m intervals is fine, but if some units collected data at 1hr intervals, I'd feel uncomfortable modeling 1m and 1hr data at the same time.

I also don't think that sampling 1 obs per minute removes autocorrelation in your dataset. Especially for a distributional model. Does your modeling approach address autocorrelation well? If not, then you will need to address this.

I really liked that you combined probability of species occurrence with that of flying within a dangerous altitude. That is a gap that has been missing in many studies.

I'm concerned that your modeling approach doesn't really seem to have a random effect for bird ID. This is problematic since your sample sizes are so different among birds. Partially this will be fixed when you explain your filtering procedure. That said, if some modeled birds had 77 data points and some had 40,000, then you have a bias issue in your analysis.

I think it would be good to explain your modeling approach a bit more clearly. The approach you took is not typical and neural networks have been critiqued in the biological literature of late (specifically in classification fields - they often perform worse at classification than do other models - the Nathan 2012 paper you cite as an example of neural networks being popular

actually shows them performing considerably worse than several other methods considered in that paper; sometimes they actually had the worst performance). So, in brief, try to more clearly explain your modeling approach. Be sure to include a description of the assumptions of the models.

Results –

Table 2 – it would be really good to include the sampling interval – max and min time between GPS fixes, for each bird. That is kind of important to note, since you combine all of them together.

I think your opening paragraph of this section can be shorter and more efficient. Try to cut the length by 33-50% but keep the same amount of information. For example, the first few sentences (lines 307-313) could be replaced with the following:

“After filtering, we retained data from 30 bearded vultures tagged in Switzerland, France, Italy or Austria and tracked from Sept 2014 to Dec 2020.”

Everything else should either be in the methods or is repetitive or not needed.

You really do need to explain your filtering approach in the Methods section.

As noted previously, I’m not comfortable with including birds with only 1 day of tracking data – unless handled very carefully, you are going to weight those birds disproportionately.

In your results, it is really important to present the figures in order of appearance. Right now Fig 3 is referenced before Fig 1 and 2. I suspect you switched things around at some point but it is confusing.

As presently shown, the risk map is not the “final” product. To me that is the most important component of your results, so I’d rather see that last.

Discussion – your first sentence is only partly correct. There are models that include a vertical dimension, there are none that I’m aware of that include both distributional models and flight behavior. That is an improvement, you should be more clear about this.

In general, I found the discussion to be a little underwhelming. To me, your most important result is the risk map. You should start with that and discuss what it means. Two things to discuss are the improvements you make to prior risk modeling and the relevance to vulture conservation. You discuss both of these but not really in a manner that is organized to highlight the really cool aspects of your work. I suggest reorganizing and trying to focus more on key messages and meanings.

You make a compelling argument in the ms that bearded vulture distribution depends strongly on the distribution of ibex. That is pretty neat but I think it is also atypical for many soaring species that interact with wind turbines at this spatial scale. Golden eagles, griffon vultures, white-tailed sea eagles, bald eagles, etc. – all of these tend to be broadly distributed across the landscape and updraft (or water) is often more important than prey distributions for these species. You might want to add a sentence or two talking about how differences in distribution might influence similar risk maps for other species. Since this is such a novel part of your work, talking about this would highlight your novelty and also characterize the general relevance to other taxa.

Tables and Figures – Table 2 needs to include the fix rate (time between fixes).

Figure 3 confused me. You have three different color scales and apparently 3 different types of information included. After reading the caption several times, I think I understood it. I think you can fix the caption by making clear that the figures show the progression of your research approach. This is not just a data figure, it is a graphical representation of your approach to analysis. Say that in your first sentence. Perhaps something like "Graphical representation of the research approach used to model risk to bearded vultures from wind turbines. Maps show the data layers used as input into models and the risk maps that are the final product of that modeling exercise." Also, your color legends need more explanation. Simply showing classes or a range from 0 to 1 doesn't help much.

I think this is a really nice paper that represents an improvement on past work. With a bit of effort, it will make a very useful contribution.

Decision letter (RSOS-211041.R0)

Dear Mr Vignali

The Editors assigned to your paper RSOS-211041 "A predictive flight-height model for avoiding future conflicts between an emblematic raptor and wind energy development in the Swiss Alps" have now received comments from reviewers and would like you to revise the paper in accordance with the reviewer comments and any comments from the Editors. Please note this decision does not guarantee eventual acceptance.

Please submit your revised manuscript and required files (see below) no later than 21 days from today's (ie 23-Jul-2021) date. Note: the ScholarOne system will 'lock' if submission of the revision is attempted 21 or more days after the deadline. If you do not think you will be able to meet this deadline please contact the editorial office immediately.

on behalf of Professor Enrico Bertuzzo (Associate Editor) and Pete Smith (Subject Editor)
openscience@royalsociety.org

Associate Editor Comments to Author (Professor Enrico Bertuzzo):

The manuscript has now been reviewed by two experts in the field. They both find that the manuscript could potentially contribute significantly to the literature. However, they both point at some issues that should be addressed in order to improve the paper and qualify it for publications. They have provided a very detailed and constructive list which I invite the authors to take in full considerations. In particular I would like to further stress few points:

- better describe the methods used and justify it (see also autocorrelation issue).
- address the issue related to the different number of datapoints per bird.
- biological vs political boundary: address it with a new analysis or otherwise justify it explicitly in the main text.

Reviewer comments to Author:

Reviewer: 1

Comments to the Author(s)

This paper builds a model from GPS tracking data from bearded vultures to predict the likelihood of a vulture flying within the rotor swept area of a turbine (<200 m agl), based on relevant topographic and environmental variables. Authors use this model to predict the probability of flight <200 m agl across the species range in the Swiss Alps. They then use a previously published predictive map of the probability of bearded vulture occurrence in the same region (Vignali et al 2021) to combine the probability of occurrence and flying in the risky zone to produce a high-risk conflict map to guide sustainable development of wind energy away from areas expected to be highly risky to this species.

I must first note in my review that I am not familiar with the modelling method - deep feedforward neural networks and thus have not commented on the section on the modelling approach. However, the model appear to have been well-cross validated, with both between birds and between region performance being tested. It is hard to tell without being familiar with the deep feedforward neural network procedures, but I would usually expect cross-validation methods to be used to inform model selection in addition to model performance, which has not been done and might be an avenue to explore.

Overall, the paper addresses an important and rapidly growing conservation dilemma and provides maps which will be useful for planning new wind energy developments in locations which pose the least collision risk to this species. This is not the first collision risk map for bearded vultures, which has already been achieved by Reid et al. 2015 [Using spatial analyses of bearded vulture movements in southern Africa to inform wind turbine placement, Journal of Applied Ecology]. Although referenced in a couple of places, there has not been much mention of this work, and a discussion on the pros/cons of the two methods or how this work is different and/or progresses conservation further would be merited.

My main concern is that the majority of the tracking data has come from juvenile, immature and subadult birds. According to Kruger et al. (2014) [Differential range use between age classes of southern African bearded vultures, PlosOne] bearded vultures only reach adulthood at 6 years old, thus only one individual was tracked as an adult in this study. Therefore, I think the authors can only really achieve a conflict map for the sub-adult sector of the population. Further more specific comments are given below.

Statistical methods

I am unable to comment in depth on the primary methodology which I am unfamiliar with – deep feedforward neural network models. It isn't clear to me if different variable combinations have been tested, or if they have all been assumed to be important predictors of flight height. For example, the permutation importance shows that 'eastness' has an importance of around 3%. Has it been tested if the model would be more parsimonious without this variable? Or is this accounted for in the neural network?

L153. "average wind speed at 100 m a.g.l. was extracted from Swiss Wind Atlas". Is this the average across the year? Derived from monthly, daily or hourly means? Can you give more information on how it was calculated and the spatial resolution of this variable? Do you think the average conditions represent the actual conditions available? E.g. an area might be very windy in one season or time of day, but not windy at all at another time. The 'average' for such an area would not actually capture either of the conditions available. If this is the case, I would recommend making predictions for actual wind scenarios that occur through the year/day and then averaging the predicted value for each location. This would also require associating each fix with the actual conditions when it was recorded for the analyses stage.

Age classes and data

Previous work by the authors which this work relies on (Vignali et al 2021), i.e. the habitat suitability model, was modelled separately for juveniles and adults, as well as summer and winter. It seems that the majority of the tracking data used in this paper comes from juvenile or subadult vultures, with only one adult being tagged and one additional bird making it to 3.5 years, and two others to 2.7 years (this was according to tracking days in table 2 – however, this is possibly 9 individuals collected data >3 years old, see below, but it is unclear which individuals or how much data). Nevertheless, following Kruger et al. (2014) bearded vultures only reach maturity at 6 years. Thus, the work here really can only assess juvenile/subadult risk and this should be made clear. As a result, I would recommend removing the limited adult tracking data from the analyses. Furthermore, the habitat suitability map used in the analyses should be an aggregation of Fig 4a and Fig 4c (from Vignali et al 2021), i.e. also not including adult data.

L320 – "Some devices (N=9) recorded data for more than 3 years. This is approximately the age at which juvenile bearded vultures change their behaviour, shifting from an exploring phase to a phase of territory establishment [40,43]."

From Table 2, only one individual had more than 1095 tracking days in the analyses. However, I understand this might be number of unique tracking days in the analyses – not days/age since tagging, which would also be useful in Table 2. Kruger et al 2014, used the following age classes: juvenile (post-fledging to 2 years), immature (2–4 years), sub-adult (4–6 years) and adult (.6 years), which is quite a big discrepancy from this paper.

Also, how do you think having different ages, from fledgling to subadult might have affected the results?

L307-310 and Table 2- One individual had only one day of tracking. Is it worth keeping this there? What happened to this individual? Is the tracking data likely to represent 'normal' behavior? Further, I see this individual is one of the ones which had lower AUC cross-validation score (L416).

Other

L361-373. Intrigued as to why these results haven't been compared with Reid et al 2015 – who found a total of 66% on non-adult bearded vulture flight <200 m agl.

L369. Rushworth and Kruger 2014 not in reference list.

L382 “the spatial distribution of potential food supply has – to the best of our knowledge - never been considered so far”. Although in a slightly different format, Reid et al 2015 included the variable “distance to vulture restaurant” which is equal to the distribution to reliable food sources for bearded vultures.

Minor

L89 Suggest removing “radio-” since it doesn't have the fine precisions that GPS tracking can achieve; particularly not altitude measurements.

L.155 I am surprised TPI and slope unevenness are not correlated – what was their correlation coefficient?

L201. Since you have a bunch of different tags which might record speed slightly differently, I would suggest using a between peaks minimum as a unique cutoff for each individual – following Schaub, et al 2020 [Collision risk of Montagu's Harriers *Circus pygargus* with wind turbines derived from high-resolution GPS tracking. *Ibis*]

L208 – one location per minute seems like autocorrelation may still be an issue?

Very minor. The paper would benefit from English proof reading, some very minor points are given here:

L24 “..with future wind turbine deployments...”. Edit plurals.

L30 Change “expansions” to “expanses”

L56. Edit needed to second half of sentence.

L87-88. “...as spatially-explicit predictive models allow extrapolation to areas...”. Grammar edits.

L92. Change ‘gate’ to ‘door’.

L103. suggest changing “new” to “growing”.

L105-106. Suggest restructuring sentence to avoid “if”. Perhaps “Using a spatial model...”?

L151. Changes ‘providers’ to ‘sources’

L225. Comma ‘,’ after ‘very large datasets’.

L287. ..., this regardless of...

L493. “believe”

Reviewer: 2

Comments to the Author(s)

This is a nice paper on risk mapping and it is an advancement over previous risk mapping. The two prior raptor-wind turbine risk maps based on flight behavior (Reid et al. 2015 (citation 33 in the ms) and Miller et al. 2014 (not cited, but I've suggested it in the text) don't do a great job of combining probability of occurrence with probability of risky flight. This paper explicitly does that and so it is a step forward.

There are a number of changes that I think would make this manuscript better. A few of the most important items here:

- I don't agree with your use of the term “flight height”. It is vogue to use it, but it is also incorrect. You note Peron's definition of “flight height” (your line 175). I appreciate that he defined this, and I appreciate that you explain how you chose your terms, but his definition leads to sloppiness. Height describes how tall something is. I've corrected it a few times in the pdf, up to you and the editors for a final decision.

- I'm curious as to why you limited your modeling to a political region (Switzerland) rather than a biologically defined region. You have lots of evidence that the birds ignore political boundaries.

For your reports to funders, sure, a political region makes sense. But for a biological study, a biological region makes more sense.

- There needs to be a much more detailed description of your data filtering process and the thresholds you use for inclusion in the analysis. You could do this in the SI (if there are space limitations in the ms) but better in the main ms if you can.

- I am concerned that your data set includes birds with only 1 day of data collection and as few as 77 data points and other birds with as many as 1,300 days and 40,000 data points. Despite this, your modeling approach does not include a random effect for bird, which means that individuals are weighted differently. If you can set a random effect for bird, that would be best. However, since it is probably not trivial to include a random effect in your models, you should probably set some threshold for inclusion and justify it, either with citations or with a simulation exercise.

- I'm concerned about autocorrelation in your dataset. I don't know the modeling techniques you describe well enough to understand if they handle autocorrelation well or not. That said, you randomly selected one obs per minute. That means the data are anywhere from 1s to 120s apart. There is still high autocorrelation in data collected at that time interval. As such, your statistical models need to handle autocorrelation. If you were using a more standard modeling approach, I could include some suggestions – certain non-parametric model types handle correlated data, or imposing a correlation structure on your models works, etc. But I really don't know what to suggest in this case – that one is up to you.

- Several spots in the ms could be substantially shorter.

- The literature cited and citations in text have some issues to address. In most places in the text you use numbers (presumably to match journal format), but in a number of places you use author names. Try to clean that up, as this got confusing in a few spots.

Please see the pdf for some minor comments and some repetition of things I've written below ("RSOS-211041_Proof_hi_review comments.pdf")

Title – I would change “height” to “altitude”

Introduction – In general, I think the introduction is pretty strong. It would be good to shorten it a bit. The last two paragraphs could be combined and shortened a bit – you don't need to go into the methods too much here, for example. You also have two paragraphs that each only have 2 sentences. Generally it is better to have at least 3 sentences in a paragraph. You can shorten and eliminate some of this text to combine. For example, the last 2 sentences of the first paragraph contain information repeated somewhat in the second sentence of the second paragraph and the sentence that starts at line 69. All that said, the intro is strong and relevant to the topic and mostly well cited.

I've put a number of notes in the pdf. A few to expand on here.

- “lift” is a force generated by the wing (the airfoil). So the words “orographic lift” are nonsensical (again, lots of people use this term, but it is incorrect). Simple fix – “orographic updraft”. Also, “thermal updraft” (not “thermal lift”).

- You suggested that we need planning tools to avoid deploying wind facilities in areas where conflicts with biodiversity will occur. Unfortunately avoiding conflict is a dream we are unlikely to realize (look at projections for expansion of wind in the coming 30 years – there will be lots of conflict). We need to talk about “minimization” of conflict, not avoiding it. Easy but important change to the ms.

Methods – I'm curious as to why you made the decision – limit the modeling exclusively to Switzerland. I get that the funding is focused on one country. That said, I have a hard time believing that your tagged birds never left the country. Clearly also, the alpine zone extends beyond the borders of this one country. This is mostly splitting hairs and is certainly not a reason

to reject the manuscript. But wouldn't it be more reasonable to model a biologically relevant region, rather than a politically relevant region? Perhaps the "contiguous Alpine areas within 500km of known nests of vultures" or something like that?

My question became more relevant when I realized that many of the birds whose data you used were tagged outside of Switzerland. I think there is good rationale for using a biologically defined study area, rather than an politically defined one.

Can you go into a bit more detail as to what is an "anthropic area" in Switzerland? In the USA, we use the term "urban" for areas in cities. But from my perspective, 90% or more of Switzerland is dominated by humans. As such, specificity seems really important when clarifying your land cover classifications.

Line 170-171 - what is the slowest rate of data collection of any GPS devices?

Rather than simply using flight speed as a threshold to remove "non-flighted" points, you should use both flight speed and altitude. A vulture at high altitudes in a thermal might be moving very quickly upwards but very slowly in a horizontal plane. See both Murgatroyd 2021 and Poessel 2018 for this. Murgatroyd says that Schaub et al. 2020 Ibis also does this.

I'm concerned that some of your birds have <10 days of data collection and that you are modeling birds with so few days of data collection in the same batch as birds with more than 1000 days of data collection. I think it would be good to set a minimum threshold for inclusion in your models. Maybe 15 or 20 days? Likewise, set a threshold for a minimum number of points. Maybe 150 or 200 data points, or at least 1 per day over your minimum time period?

OK - having read the results, you say that you had "requirements for being included in the analysis". I missed those in the methods. Can you clarify in the methods what were the requirements for being included in the analysis? I suspect that will address my concerns above. Likewise, be sure to explain whether your filtering of GPS points was before or after you filtered birds.

I don't love your resampling down to 1m data. The resampling itself is fine. However, the bigger problem is that you don't clearly explain the slowest data rate your transmitters collected. 1m intervals is fine, but if some units collected data at 1hr intervals, I'd feel uncomfortable modeling 1m and 1hr data at the same time.

I also don't think that sampling 1 obs per minute removes autocorrelation in your dataset. Especially for a distributional model. Does your modeling approach address autocorrelation well? If not, then you will need to address this.

I really liked that you combined probability of species occurrence with that of flying within a dangerous altitude. That is a gap that has been missing in many studies.

I'm concerned that your modeling approach doesn't really seem to have a random effect for bird ID. This is problematic since your sample sizes are so different among birds. Partially this will be fixed when you explain your filtering procedure. That said, if some modeled birds had 77 data points and some had 40,000, then you have a bias issue in your analysis.

I think it would be good to explain your modeling approach a bit more clearly. The approach you took is not typical and neural networks have been critiqued in the biological literature of late (specifically in classification fields - they often perform worse at classification than do other models - the Nathan 2012 paper you cite as an example of neural networks being popular

actually shows them performing considerably worse than several other methods considered in that paper; sometimes they actually had the worst performance). So, in brief, try to more clearly explain your modeling approach. Be sure to include a description of the assumptions of the models.

Results –

Table 2 – it would be really good to include the sampling interval – max and min time between GPS fixes, for each bird. That is kind of important to note, since you combine all of them together.

I think your opening paragraph of this section can be shorter and more efficient. Try to cut the length by 33-50% but keep the same amount of information. For example, the first few sentences (lines 307-313) could be replaced with the following:

“After filtering, we retained data from 30 bearded vultures tagged in Switzerland, France, Italy or Austria and tracked from Sept 2014 to Dec 2020.”

Everything else should either be in the methods or is repetitive or not needed.

You really do need to explain your filtering approach in the Methods section.

As noted previously, I’m not comfortable with including birds with only 1 day of tracking data – unless handled very carefully, you are going to weight those birds disproportionately.

In your results, it is really important to present the figures in order of appearance. Right now Fig 3 is referenced before Fig 1 and 2. I suspect you switched things around at some point but it is confusing.

As presently shown, the risk map is not the “final” product. To me that is the most important component of your results, so I’d rather see that last.

Discussion – your first sentence is only partly correct. There are models that include a vertical dimension, there are none that I’m aware of that include both distributional models and flight behavior. That is an improvement, you should be more clear about this.

In general, I found the discussion to be a little underwhelming. To me, your most important result is the risk map. You should start with that and discuss what it means. Two things to discuss are the improvements you make to prior risk modeling and the relevance to vulture conservation. You discuss both of these but not really in a manner that is organized to highlight the really cool aspects of your work. I suggest reorganizing and trying to focus more on key messages and meanings.

You make a compelling argument in the ms that bearded vulture distribution depends strongly on the distribution of ibex. That is pretty neat but I think it is also atypical for many soaring species that interact with wind turbines at this spatial scale. Golden eagles, griffon vultures, white-tailed sea eagles, bald eagles, etc. – all of these tend to be broadly distributed across the landscape and updraft (or water) is often more important than prey distributions for these species. You might want to add a sentence or two talking about how differences in distribution might influence similar risk maps for other species. Since this is such a novel part of your work, talking about this would highlight your novelty and also characterize the general relevance to other taxa.

Tables and Figures – Table 2 needs to include the fix rate (time between fixes).

Figure 3 confused me. You have three different color scales and apparently 3 different types of information included. After reading the caption several times, I think I understood it. I think you can fix the caption by making clear that the figures show the progression of your research approach. This is not just a data figure, it is a graphical representation of your approach to analysis. Say that in your first sentence. Perhaps something like "Graphical representation of the research approach used to model risk to bearded vultures from wind turbines. Maps show the data layers used as input into models and the risk maps that are the final product of that modeling exercise." Also, your color legends need more explanation. Simply showing classes or a range from 0 to 1 doesn't help much.

I think this is a really nice paper that represents an improvement on past work. With a bit of effort, it will make a very useful contribution.

===PREPARING YOUR MANUSCRIPT===

Your revised paper should include the changes requested by the referees and Editors of your manuscript. You should provide two versions of this manuscript and both versions must be provided in an editable format:
 one version identifying all the changes that have been made (for instance, in coloured highlight, in bold text, or tracked changes);
 a 'clean' version of the new manuscript that incorporates the changes made, but does not highlight them. This version will be used for typesetting if your manuscript is accepted.

===PREPARING YOUR REVISION IN SCHOLARONE===

Author's Response to Decision Letter for (RSOS-211041.R0)

See Appendix B.

RSOS-211041.R1 (Revision)

Review form: Reviewer 2

Is the manuscript scientifically sound in its present form?

Yes

Are the interpretations and conclusions justified by the results?

Yes

Is the language acceptable?

Yes

Do you have any ethical concerns with this paper?

No

Have you any concerns about statistical analyses in this paper?

No

Recommendation?

Accept with minor revision (please list in comments)

Comments to the Author(s)

Thanks for the careful reply to the reviewer comments. I have many fewer suggestions on this version of the manuscript.

Two brief comments on your response to reviewers.

First, in response to a comment from reviewer 1 about wind measurements at a specific site – you say “However, the only way to do this would have been using measurements from the weather stations that are closes to the flight locations...”. That is not quite correct. There are R packages (RNCEP) and web applications (MoveBank) that can use modeled data to interpolate to a GPS location and date/time. They are flawed in the same way as you note for weather stations, but they are available. The more pertinent response to this comment though is that by using an “average” dataset, you enable prediction much more effectively. It isn’t much good to predict to specific wind conditions, since you never know when those specific conditions will occur. Much better to predict based on an average, then you can understand the future better. I don’t think you need to change anything, I bring this up mostly for clarification.

Second, in my first review I noted that there needed to be a better description of the filtering process. You then went into a bit of detail on filtering for birds. I fear that I was not clear in my description of what I meant by filtering. Presumably you also did some filtering for GPS errors. For example, it is not uncommon to see 3 GPS fixes at 100m in altitude, then a 4th at 2000m, then three more at 100m. In that case the one in the middle is probably a GPS error – especially if time intervals are short. We see this frequently, no matter who the manufacturer. There are also locational errors occasionally – for example, a colleague in Australia sometimes gets GPS fixes in Africa. One out of every 1,000 or 10,000 GPS data points, but it happens. Did you do any filtering for this type of GPS error?

Manuscript –

Your abstract is mostly introduction and methods. You have only one sentence of “results”. Generally, it is better if an abstract is about 25% introduction, 15-25% methods, 40% results, and 10-20% discussion/conclusions. It would be good to insert more results into your abstract.

Introduction –

Line 52 – remove “endangered or threatened” and replace with “of conservation concern”. Endangered and threatened have very specific (sometimes legal) meanings, and we are often concerned about species that don’t meet that threshold.

The introduction is very nice – concise, to the point.

Methods –

It would be good to clarify the data filtering you did, as I note above.

You should clarify the inter-fix interval for the transmitters. Not only how many fixes per day (which you report in Table 2, I think), but also the time between fixes (minimum and maximum for each unit, this can also be in Table 2).

I like the explanation for modeling just in Switzerland, thank you.

Line 155 – I would change “bones” to “food” – bearded vultures will eat meat too, if available.

Line 183 – instead of “Poessel et al”, should this be (53)?

Line 205 – since this is an “and”, there really isn’t a “Second criterion”, right? It is just one criteria with two parts? A pedantic point, but you might rephrase to say “Thresholding in this manner may have removed some valid flying positions....”

Line 176 and 210 – If I read this correctly, for the telemetry units that collected data at long time intervals (more than 1 minute), you did not resample to 1 minute, you just used the data as they came, correct? So data analyzed were as fast as 1 fix/minute and as slow as about 1 fix every 15 minutes, correct? You should clarify this for the reader.

Line 227 – change “it has” to “they have”

Line 233 – change “heights in” to “heights of”

Line 237 – is a 0 for a location outside of the critical altitude range? If so, perhaps say that.

Line 252 – delete extra space between “30” and “%”

Line 255 – is “epochs” the correct word here? I know what it means, but I am not familiar with its use in this context. (also used in line 284)

Line 265 – perhaps change to “we used 10 km blocks, to verify the ability....”

Results –

Table 2 – this might be better as an SI table. Also, “fix rate” needs to be a rate, not a count. Is it “per day”? or something else? Can you also report the “inter-fix interval” – which is sometimes 1Hz and sometimes a lot slower?

Line 324 – “after applying the subsampling procedure” – this is fine but you don’t call it “the subsampling procedure” in the methods. This is the first time you’ve used this terminology. You should name it in the methods and then use that name here. Minor point, easy to fix.

Line 345 – 347 – this is interesting, as wind speed is often correlated with flight altitude. It is not a linear relationship though – at low wind speeds, birds are at low altitudes, at moderate wind speeds birds are at higher altitudes, and at very high wind speeds, birds are at low altitudes. Lanzone et al. 2012 shows some of this, as does Bonner et al. 2021 (online early). Does your modeling approach allow for non-linear relationships in your data?

Lanzone, M., T. Miller, P. Turk, D. Brandes, C. Halverson, C. Maisonneuve, J. Tremblay, J. Cooper, K. O’Malley, R. Brooks, & T. Katzner. 2012. Flight responses by a migratory soaring raptor to changing meteorological conditions. *Biology Letters*, 8:710-713.

Bonner, S.R., S.A. Poessel, J.C. Brandt, M.T. Astell, J.R. Belthoff, and T.E. Katzner. Drivers of flight performance of California condors. *Journal of Raptor Research*. In press.

Nothing for you to change here, but you may want to explore this in the discussion.

Figure 2 – your axis labels are inconsistent. The plot for slope does not have 1.00, the others do.

Figure 3 – I really like this one and the new caption is helpful. Can you make the country borders black instead of grey? They are hard to see. Also, since your discussion is mostly about the “Swiss Alpine Range”, you probably should clarify where that is on the map.

Discussion –

Line 370 – also not surprising since steep slopes generate orographic updraft. Orographic updraft tends to subside at about 200m above ground, and so birds can’t get high. When they use thermals, they can get much higher. See Katzner et al. 2012 for an explanation of this (citation 55 in the ms). (ok, you get into this a bit in line 393, although you don’t explicitly say that orographic updrafts keep birds at lower altitudes, you may want to say this).

Line 383 – change “exposition” to “exposure”, also change the end of the sentence to “explaining the formation of thermal and orographic updraft”

Your AUC is 0.7, you might also want to report the “lift” – some metric that compares your predictions to random. I think your lift is about 30%, but not sure.

Line 480 – 483 – not sure what this sentence is trying to say. Perhaps rephrase this for clarity.

Line 483 – 485 – perhaps rewrite this entire sentence, “Although we adopted a conservative approach, our results suggest that birds fly above the critical altitude range in about 9.1% (2,351 km²) of the areas previously classified as having high conflict potential.”

Line 486 – 496 – I’m not worried about the age issues, as was the other reviewer. In my experience, if a bird is in a certain spot, it will respond to the environment in a set way, regardless of age. What differs about flight behavior is ranging behavior - things like home range size and degree of wandering. But flight altitude is fairly constant, regardless of age. Citations for soaring birds that confirm this are:

This paper shows age- and season-related difference in ranging behavior:

Miller, T.A., R.P. Brooks, M.J. Lanzone, J. Cooper, K. O'Malley, D. Brandes, A. Duerr & T.E. Katzner. 2017. Space use and home range characteristics of Golden Eagles (*Aquila chrysaetos*) in eastern North America during breeding season and winter. *The Condor*. 119: 697-719.

These two papers show no age effect for flight behavior:

Lanzone et al. 2012 (as cited above)

Katzner, T.E., P.J. Turk, A.E. Duerr, T.A. Miller, M.J. Lanzone, J.L. Cooper, D. Brandes, J.A.

Tremblay & J. Lemaître. 2015. Use of multiple modes of flight subsidy by a soaring terrestrial bird, the golden eagle *Aquila chrysaetos*, when on migration. *Journal of the Royal Society Interface*. 12: 20150530.

Lines 434 – 510 – to me, this section is a bit long and it doesn't really get to the biology. It is methodological and focuses a little too much on weaknesses. Every paper has weaknesses, science is incremental, and each new iteration improves on the prior one. That is normal. I think it is fine to mention these things and you can definitely leave this as it is. However, if the editors tell you to make the paper shorter, this is the section you should shorten. You can probably cut it in half, while bringing up the same points and making your paper more impactful.

Nice job with this paper, I think it will be influential.

Decision letter (RSOS-211041.R1)

Dear Mr Vignali

On behalf of the Editors, we are pleased to inform you that your Manuscript RSOS-211041.R1 "A predictive flight-altitude model for avoiding future conflicts between an emblematic raptor and wind energy development in the Swiss Alps" has been accepted for publication in Royal Society Open Science subject to minor revision in accordance with the referees' reports. Please find the referees' comments along with any feedback from the Editors below my signature.

Please submit your revised manuscript and required files (see below) no later than 7 days from today's (ie 05-Jan-2022) date. Note: the ScholarOne system will 'lock' if submission of the revision is attempted 7 or more days after the deadline. If you do not think you will be able to meet this deadline please contact the editorial office immediately.

Please note article processing charges apply to papers accepted for publication in Royal Society Open Science (<https://royalsocietypublishing.org/rsos/charges>). Charges will also apply to papers transferred to the journal from other Royal Society Publishing journals, as well as papers submitted as part of our collaboration with the Royal Society of Chemistry

(<https://royalsocietypublishing.org/rsos/chemistry>). Fee waivers are available but must be requested when you submit your revision (<https://royalsocietypublishing.org/rsos/waivers>).

on behalf of Professor Enrico Bertuzzo (Associate Editor) and Pete Smith (Subject Editor)
openscience@royalsociety.org

Associate Editor Comments to Author (Professor Enrico Bertuzzo):

Comments to the Author:

The manuscript was reviewed by Reviewer #2 who initially suggested a major revision and, as the authors can read, they are now satisfied with the revision. I carefully checked how the authors responded to the comments raised by reviewer #1 and also in this case I do not see any further revision required.

The manuscript can therefore be accepted for publication pending a minor revision to address the residual minor comments from the Reviewer.

Reviewer comments to Author:

Reviewer: 2

Comments to the Author(s)

Thanks for the careful reply to the reviewer comments. I have many fewer suggestions on this version of the manuscript.

Two brief comments on your response to reviewers.

First, in response to a comment from reviewer 1 about wind measurements at a specific site – you say “However, the only way to do this would have been using measurements from the weather stations that are closest to the flight locations...”. That is not quite correct. There are R packages (RNCEP) and web applications (MoveBank) that can use modeled data to interpolate to a GPS location and date/time. They are flawed in the same way as you note for weather stations, but they are available. The more pertinent response to this comment though is that by using an “average” dataset, you enable prediction much more effectively. It isn’t much good to predict to specific wind conditions, since you never know when those specific conditions will occur. Much better to predict based on an average, then you can understand the future better. I don’t think you need to change anything, I bring this up mostly for clarification.

Second, in my first review I noted that there needed to be a better description of the filtering process. You then went into a bit of detail on filtering for birds. I fear that I was not clear in my description of what I meant by filtering. Presumably you also did some filtering for GPS errors. For example, it is not uncommon to see 3 GPS fixes at 100m in altitude, then a 4th at 2000m, then three more at 100m. In that case the one in the middle is probably a GPS error – especially if time intervals are short. We see this frequently, no matter who the manufacturer. There are also locational errors occasionally – for example, a colleague in Australia sometimes gets GPS fixes in Africa. One out of every 1,000 or 10,000 GPS data points, but it happens. Did you do any filtering for this type of GPS error?

Manuscript –

Your abstract is mostly introduction and methods. You have only one sentence of “results”. Generally, it is better if an abstract is about 25% introduction, 15-25% methods, 40% results, and 10-20% discussion/conclusions. It would be good to insert more results into your abstract.

Introduction –

Line 52 – remove “endangered or threatened” and replace with “of conservation concern”. Endangered and threatened have very specific (sometimes legal) meanings, and we are often concerned about species that don’t meet that threshold.

The introduction is very nice – concise, to the point.

Methods –

It would be good to clarify the data filtering you did, as I note above.

You should clarify the inter-fix interval for the transmitters. Not only how many fixes per day (which you report in Table 2, I think), but also the time between fixes (minimum and maximum for each unit, this can also be in Table 2).

I like the explanation for modeling just in Switzerland, thank you.

Line 155 – I would change “bones” to “food” – bearded vultures will eat meat too, if available.

Line 183 – instead of “Poessel et al”, should this be (53)?

Line 205 – since this is an “and”, there really isn’t a “Second criterion”, right? It is just one criteria with two parts? A pedantic point, but you might rephrase to say “Thresholding in this manner may have removed some valid flying positions....”

Line 176 and 210 – If I read this correctly, for the telemetry units that collected data at long time intervals (more than 1 minute), you did not resample to 1 minute, you just used the data as they came, correct? So data analyzed were as fast as 1 fix/minute and as slow as about 1 fix every 15 minutes, correct? You should clarify this for the reader.

Line 227 – change “it has” to “they have”

Line 233 – change “heights in” to “heights of”

Line 237 – is a 0 for a location outside of the critical altitude range? If so, perhaps say that.

Line 252 – delete extra space between “30” and “%”

Line 255 – is “epochs” the correct word here? I know what it means, but I am not familiar with its use in this context. (also used in line 284)

Line 265 – perhaps change to “we used 10 km blocks, to verify the ability....”

Results –

Table 2 – this might be better as an SI table. Also, “fix rate” needs to be a rate, not a count. Is it “per day”? or something else? Can you also report the “inter-fix interval” – which is sometimes 1Hz and sometimes a lot slower?

Line 324 – “after applying the subsampling procedure” – this is fine but you don’t call it “the subsampling procedure” in the methods. This is the first time you’ve used this terminology. You should name it in the methods and then use that name here. Minor point, easy to fix.

Line 345 – 347 – this is interesting, as wind speed is often correlated with flight altitude. It is not a linear relationship though – at low wind speeds, birds are at low altitudes, at moderate wind speeds birds are at higher altitudes, and at very high wind speeds, birds are at low altitudes. Lanzone et al. 2012 shows some of this, as does Bonner et al. 2021 (online early). Does your modeling approach allow for non-linear relationships in your data?

Lanzone, M., T. Miller, P. Turk, D. Brandes, C. Halverson, C. Maisonneuve, J. Tremblay, J. Cooper, K. O’Malley, R. Brooks, & T. Katzner. 2012. Flight responses by a migratory soaring raptor to changing meteorological conditions. *Biology Letters*, 8:710-713.

Bonner, S.R., S.A. Poessel, J.C. Brandt, M.T. Astell, J.R. Belthoff, and T.E. Katzner. Drivers of flight performance of California condors. *Journal of Raptor Research*. In press.

Nothing for you to change here, but you may want to explore this in the discussion.

Figure 2 – your axis labels are inconsistent. The plot for slope does not have 1.00, the others do.

Figure 3 – I really like this one and the new caption is helpful. Can you make the country borders black instead of grey? They are hard to see. Also, since your discussion is mostly about the “Swiss Alpine Range”, you probably should clarify where that is on the map.

Discussion –

Line 370 – also not surprising since steep slopes generate orographic updraft. Orographic updraft tends to subside at about 200m above ground, and so birds can’t get high. When they use thermals, they can get much higher. See Katzner et al. 2012 for an explanation of this (citation 55 in the ms). (ok, you get into this a bit in line 393, although you don’t explicitly say that orographic updrafts keep birds at lower altitudes, you may want to say this).

Line 383 – change “exposition” to “exposure”, also change the end of the sentence to “explaining the formation of thermal and orographic updraft”

Your AUC is 0.7, you might also want to report the “lift” – some metric that compares your predictions to random. I think your lift is about 30%, but not sure.

Line 480 – 483 – not sure what this sentence is trying to say. Perhaps rephrase this for clarity.

Line 483 – 485 – perhaps rewrite this entire sentence, “Although we adopted a conservative approach, our results suggest that birds fly above the critical altitude range in about 9.1% (2,351 km²) of the areas previously classified as having high conflict potential.”

Line 486 – 496 – I’m not worried about the age issues, as was the other reviewer. In my experience, if a bird is in a certain spot, it will respond to the environment in a set way, regardless of age. What differs about flight behavior is ranging behavior - things like home range size and degree of wandering. But flight altitude is fairly constant, regardless of age. Citations for soaring birds that confirm this are:

This paper shows age- and season-related difference in ranging behavior:

Miller, T.A., R.P. Brooks, M.J. Lanzone, J. Cooper, K. O'Malley, D. Brandes, A. Duerr & T.E. Katzner. 2017. Space use and home range characteristics of Golden Eagles (*Aquila chrysaetos*) in eastern North America during breeding season and winter. *The Condor*. 119: 697-719.

These two papers show no age effect for flight behavior:

Lanzone et al. 2012 (as cited above)

Katzner, T.E., P.J. Turk, A.E. Duerr, T.A. Miller, M.J. Lanzone, J.L. Cooper, D. Brandes, J.A. Tremblay & J. Lemaître. 2015. Use of multiple modes of flight subsidy by a soaring terrestrial bird, the golden eagle *Aquila chrysaetos*, when on migration. *Journal of the Royal Society Interface*. 12: 20150530.

Lines 434 – 510 – to me, this section is a bit long and it doesn't really get to the biology. It is methodological and focuses a little too much on weaknesses. Every paper has weaknesses, science is incremental, and each new iteration improves on the prior one. That is normal. I think it is fine to mention these things and you can definitely leave this as it is. However, if the editors tell you to make the paper shorter, this is the section you should shorten. You can probably cut it in half, while bringing up the same points and making your paper more impactful.

Nice job with this paper, I think it will be influential.

===PREPARING YOUR MANUSCRIPT===

one version should clearly identify all the changes that have been made (for instance, in coloured highlight, in bold text, or tracked changes);

a 'clean' version of the new manuscript that incorporates the changes made, but does not highlight them. This version will be used for typesetting

===PREPARING YOUR REVISION IN SCHOLARONE===

-- If you are requesting an article processing charge waiver, you must select the relevant waiver option (if requesting a discretionary waiver, the form should have been uploaded, see 'File upload' above).

-- If you have uploaded any electronic supplementary (ESM) files, please ensure you follow the guidance at <https://royalsociety.org/journals/authors/author-guidelines/#supplementary-material> to include a suitable title and informative caption. An example of appropriate titling and captioning may be found at https://figshare.com/articles/Table_S2_from_Is_there_a_trade-off_between_peak_performance_and_performance_breadth_across_temperatures_for_aerobic_scope_in_teleost_fishes_/3843624.

Author's Response to Decision Letter for (RSOS-211041.R1)

See Appendix C.

Decision letter (RSOS-211041.R2)

Dear Mr Vignali,

I am pleased to inform you that your manuscript entitled "A predictive flight-altitude model for avoiding future conflicts between an emblematic raptor and wind energy development in the Swiss Alps" is now accepted for publication in Royal Society Open Science.

on behalf of Professor Enrico Bertuzzo (Associate Editor) and Pete Smith (Subject Editor)
openscience@royalsociety.org

Appendix A**ROYAL SOCIETY
OPEN SCIENCE****A predictive flight-height model for avoiding future conflicts
between an emblematic raptor and wind energy
development in the Swiss Alps**

Journal:	Royal Society Open Science
Manuscript ID	RSOS-211041
Article Type:	Research
Date Submitted by the Author:	23-Jun-2021
Complete List of Authors:	Vignali, Sergio; University of Bern Loercher, Franziska; Stiftung Pro Bartgeier; SWILD Stadtökologie Wildtierforschung Kommunikation; Vulture Conservation Foundation Hegglin, Daniel; Stiftung Pro Bartgeier; SWILD Stadtökologie Wildtierforschung Kommunikation Arlettaz, Raphael; University of Bern, Conservation Biology Braunisch, Veronika; University of Bern; Forest Research Institute Baden-Wuerttemberg
Subject:	ecology < BIOLOGY
Keywords:	wind energy, risk mitigation, wildlife-human conflicts, spatial planning, predictive modelling, vulture conservation
Subject Category:	Ecology, Conservation, and Global Change Biology

Author-supplied statements

Relevant information will appear here if provided.

Ethics

Does your article include research that required ethical approval or permits?:

Yes

Statement (if applicable):

All the work has been conducted in accordance with relevant national and international guidelines and conforms to the legal requirements. Birds reintroduced in the Alps hatched within an international captive-breeding program (EAZA Ex situ programmes) and have been tagged before the release. In the case of wild individuals (nine birds from France and one bird from Italy), captures have been carried out in compliance with the Ethical Principles in Animal Research. All animals have not been marked for scientific reasons but for the monitoring program of the ongoing reintroduction of bearded vultures in the Alps. The GPS data have been provided to the authors by Asters Conservatoire d'espaces naturels de Haute-Savoie (FR), the Parc national de la Vanoise (FR), the National Park Hohe Tauern (AT), the Parco Nazionale dello Stelvio (IT) and the Swiss Foundation for Bearded Vultures (CH). Data from France, Austria and Italy are purely third-party data. The authors DH and FL are involved in the monitoring program of the Swiss reintroduction project. However, according to Swiss law projects involving wild animals that do not meet the definition of animal experiments in Article 3 letter c of the Animal Protection Act are not subject to any authorization procedure under animal protection law and measures that serve the protection and management of wild animals do not require an animal experiment permit. Thus, protocols, amendments, and other resources have been done according to the national guidelines.

Data

It is a condition of publication that data, code and materials supporting your paper are made publicly available. Does your paper present new data?:

Yes

Statement (if applicable):

Data including bird id, value of the predictors, and fold id for the spatial fold cross validation are included in file "bv-data.csv.gz".

The R code to reproduce the analysis is included in file "R-script.R".

Both files are compressed within the folder available at the following link:

<https://drive.google.com/file/d/1k-aNHe2HzwtFwMtBmcbhjpsp99vGHgyw/view?usp=sharing>

Conflict of interest

I/We declare we have no competing interests

Statement (if applicable):

CUST_STATE_CONFLICT :No data available.

Authors' contributions

This paper has multiple authors and our individual contributions were as below

*Statement (if applicable):*

S.V., V.B. and R.A. conceived and designed the study; S.V. carried out the analysis and wrote the first
draft of the manuscript; D.H. and F.L. organised tracking data collection and provision and provided
information on the species; R.A. initiated the research programme, provided the resources, and
coordinated the study with V.B. All authors contributed to subsequent drafts and gave final approval
for publication, with V.B. and R.A. thoroughly editing the article

**A predictive flight-height model for avoiding future conflicts**
**between an emblematic raptor and wind energy development**
**in the Swiss Alps**

Sergio Vignali^{1*}, Franziska Loercher^{2,3,4}, Daniel Hegglin^{2,3}, Raphaël Arlettaz¹, Veronika
Braunisch^{1,5}

¹Division of Conservation Biology, Institute of Ecology and Evolution, University of Bern,
Baltzerstrasse 6, CH-3012 Bern, Switzerland

² Stiftung Pro Bartgeier, Wuhrstrasse 12, CH-8003 Zurich

³ SWILD, Wuhrstrasse 12, CH-8003 Zuerich

⁴ Vulture Conservation Foundation, Wuhrstrasse 12, CH-8003 Zurich

⁵Forest Research Institute of Baden-Wuerttemberg, Wonnhaldestrasse 4, D-79100,
Freiburg, Germany

*Corresponding author: sergio.vignali@iee.unibe.ch ORCID 0000-0002-3390-5442

Abstract

The deployment of wind energy should contribute to the societal shift towards a massive
reduction of greenhouse gas emissions. Yet, wind energy and large birds, notably
soaring raptors, both depend on suitable wind conditions. Conflicts in airspace use may
thus arise between wind energy development and wildlife protection due to the risks of
collisions of birds with the blades of wind turbines. Using locations of GPS-tagged
bearded vultures, a rare scavenging raptor reintroduced into the Alps, we built a
spatially-explicit model to predict potential areas of conflict with future wind turbines
deployment in the Swiss Alps. We modelled the probability of bearded vultures flying
within the range of rotor-swept heights of wind turbines as a function of wind and
environmental conditions, including food supply (wild ungulates presence). Flight activity
within the blade-swept heights of wind turbines was generally high, concentrating on
south-exposed mountainsides, especially in areas where ibex carcasses have a high
occurrence probability, with critical areas covering vast expansions throughout the
Swiss Alps. Our model provides a spatially-explicit decision tool that will guide
authorities and energy companies for planning the deployment of wind farms without
jeopardising the chances of survival of emblematic Alpine wildlife.

Keywords

[revised manuscript text omitted]

population sizes of potentially affected species [36]. The second method equates areas
of species presence with areas of potential conflict; not accounting for actual fine-
grained species-habitat associations it remains coarse but can be valuable for
identifying broad areas of potential conflicts. The third method is the most sophisticated
and also the most informative one, as spatially-explicit predictive models allows
extrapolating to areas for which data about species presence may be deficient.
Moreover, when relying on individual-based data such as radio- or GPS-tracking, it
enables delineating areas of potential conflict with an unprecedented precision, most
notably when providing information about the height above ground at which birds fly.
This approach opens the gate towards 3D spatial modelling aimed to mitigate if not
avoid conflicts between flying vertebrates and future wind facilities development.
The aim of this study was to predict areas of the Swiss Alps where bearded vultures
(*Gypaetus barbatus*) are likely to fly within the critical height range that is typically swept
by the blades of modern wind turbines. The bearded vulture is a long-lived scavenger
listed as vulnerable in Europe [37]. It is still critically endangered in Switzerland [38].
Extirpated from many European countries in the early twentieth century [39], the
species has been reintroduced into the Alps since the 1980s, with a steadily growing
population that progressively recolonises its former historical range [40]. Several cases
of collisions (including fatalities) with ~~aerial~~ anthropogenic structures have been
reported in this re-established population [41,42], including with wind turbines which
may represent a new major source of hazard into the future [43]. In effect, Schaub et al.
(2009) have shown that even a slight increase in mortality would push the Alpine
population of bearded vultures below demographic self-sustainability. If a spatial model

we have recently developed predicts species' potential distribution, including future
expansion across the Swiss Alps [43], the present model adds a vertical dimension to
these projections. In effect, such a model would refine the prediction of potential
conflicts, as the actual use of the airspace, i.e. the flight height with respect to the blade-
swept range is accounted for.

Starting from a large dataset of GPS locations collected from tagged individuals, we
thus modelled the probability that bearded vultures fly within the critical, blade-swept
height range of wind turbines and identified the environmental and topographic
variables that drive flight height selection. The model was projected to the entire Swiss
Alpine range and combined with the previously modelled potential distribution of the
species [43] in order to show the joint probability of bearded vultures flying at risky
heights within suitable habitat. The resulting map provides useful spatial information to
delineate areas where the species would be at risk of colliding with wind turbine blades
and therefore represents a useful decision tool for planning the deployment of wind
power plants across the Swiss Alpine range while minimising their potential impacts on
emblematic biodiversity.

43 44 123 **2. Methods**

45 46 47 124 2.1 Study area and environmental variables

We modelled the flight ~~height~~ of bearded vulture across the entire Swiss Alpine range,
defined as four of the six biogeographical regions of Switzerland [44]: Northern Alps,
Inner Western Alps, Inner Eastern Alps, and Southern Alps. We used environmental

variables that represent land cover characteristics, geology, topography, food
availability and wind conditions (Table 1). Land cover information was extracted from
the digital cartographic model of Switzerland (Vector25,
<https://www.swisstopo.admin.ch/en/geodata/maps/smv/smv25.html>). This vector layer
was converted into a raster map with 25 m spatial resolution and reclassified to
represent the following ten classes: orchards, forest, bush, scree, anthropic areas,
marshland, water, rock, glacier, and remaining areas not included in the other classes
(Table S1). The geological features were derived from the simplified geotechnical map
of Switzerland which was provided as digitised vector map by the University of Bern
(<https://biblio.unibe.ch/maps/bis/publications/dl-oef21.html>) and represents the types of
the topmost rock strata ([https://data.geo.admin.ch/ch.swisstopo.geologie-geotechnik-
\[gk200/\]\(https://data.geo.admin.ch/ch.swisstopo.geologie-geotechnik-gk200/\) \[45\]\). The shapefile was converted into a raster map with 25 m spatial resolution
and reclassified into four classes: areas dominated by limestone, granite, gneiss, and
remaining geological substrates \(Table S2\). Topography was characterised with five
raster layers extracted from a digital elevation model with a spatial resolution of 25 m
\(DHM25, <https://www.swisstopo.admin.ch/en/geodata/height/dhm25.html>\). The aspect
of the study area was represented by the deviation from east and north \(*sine* and *cosine*
of aspect, respectively\). Terrain characteristics were incorporated by using the
Topographic Position Index \(TPI, Wilson, 1984\) and the slope unevenness, which
describe the elevation or slope of a cell relative to the surrounding terrain, respectively
\(both calculated within a moving window of nine pixels\). Northness and eastness were
calculated with ArcGIS 10.2, TPI and slope unevenness were derived using the *raster*
package in R \[47\]. Food availability was described using the modelled probability of
60](https://data.geo.admin.ch/ch.swisstopo.geologie-geotechnik-gk200/)

chamois and ibex occurrence, the two main providers of bones for bearded vultures,
which thus served as a proxy for food supply (for methodological details see Vignali,
Lörcher, Heggin, Arlettaz, & Braunisch, 2021, Appendix A). Finally, average wind speed
at 100 m a.g.l. was extracted from Swiss Wind Atlas [48]. Pairwise Spearman's
correlations between all continuous environmental variables were $|r_s| < 0.6$, calculated
based on 10,000 random locations. Categorical variables (i.e. land cover and geology)
were one-hot encoded while continuous variables were normalised using the mean and
standard deviation derived from the training dataset.

2.2 Species data and data processing

Between 2005 and 2020, as part of the Alpine reintroduction programme, 97 bearded
vultures have been equipped with GPS loggers (battery or solar-powered) fitted with a
leg loop harness [49]. All birds but one were tagged as fledglings, 81 thereof captive-
bred and 16 wild-hatched. In addition, one adult bird, released in 1999, was tagged in
2017 after recapture, rehabilitation and re-release. Loggers from different manufacturers
and relying on various power sources were deployed whilst GPS locations were
collected with a very heterogenous schedule. For example, some devices were
programmed to collect bursts with high frequency resolution (1 Hz) as long as the bird
was moving and the battery was sufficiently charged. Others collected GPS locations at
1-min resolution under similar conditions, while some devices recorded data with even
lower temporal resolution. Since we were interested in modelling the flight height above
ground level, we selected only data collected by GPS devices that simultaneously
recorded information on both flight altitude and instantaneous ground speed so that

non-flight locations could be excluded from the analysis (see below). Following the
definition of Péron et al. (2020) we define “flight height” as the distance between the bird
and the ground level, and “flight altitude” as the distance relative to a reference surface

[revised manuscript text omitted]

44
45

46 304

48 305 **3. Results**

51 306 3.1 Tracking data

Among all bearded vultures GPS-tagged by the Alpine reintroduction programme, 32
individuals fulfilled the requirements for being included in the analysis. Two out of these
32 individuals also had to be discarded because they yielded only five and nine GPS
locations in Switzerland, respectively. The retained 30 bearded vultures had been
tagged in all four countries involved in the Alpine reintroduction project (i.e. Switzerland,
France, Italy, and Austria), with the data used for the analysis having been collected
from September 2014 to the end of December 2020. The number of collected locations,
as well as the amount of tracking days, varied significantly among tagged individuals,
with larger sample sizes in birds released within Switzerland, and lower sample sizes in
birds that only occasionally visited the study area, stemming from release sites in the
neighbouring countries (Table 2). The number of tracking days within the Swiss Alpine
range varied from 1 to 1308 per individual while the duration of the tracking period per
individual varied according to the lifetime of the solar-battery system, any device loss or
deficiency, or in case of a bird's death. ~~Some devices (N=9) recorded data for more~~
~~321 than 3 years. This is approximately the age at which juvenile bearded vultures change~~
~~322 their behaviour, shifting from an exploring phase to a phase of territory establishment~~
323 [40,43]. A total of 2,939,411 GPS locations were retained after data cleaning, of which
77.5% were collected below 200 m a.g.l. (average proportions varying between
individuals from 58.6% to 92.1%). **After applying the above subsampling procedure,**
**326 flight height was finally modelled based on 227,313 GPS locations.**

3.2 Model architecture and predictions

The best model configuration identified during the grid search experiment was a deep
feedforward neural network with 256 units in the first hidden layer, a dropout rate of
60%, and 32 units in the second hidden layer. This model had an AUC value of 0.726
for the training dataset and 0.719 for the validation dataset. Overall, the model was able
to generalise well across birds, which was indicated by a mean training and testing AUC
of 0.716 (SD=0.003) and 0.721 (SD=0.045), respectively (Table S3), which was
comparable to the performance of the model trained using all birds. Similarly, the model
showed a good ability to generalise across different regions of the study area with a
mean training and testing AUC of 0.719 (SD=0.005) and 0.710 (SD=0.016), respectively
(Table S4).

The combination of the potential conflict map (Fig. 3d) with binary representation of the
probability of flying within the critical height below 200 m a.g.l. (Fig. 3b) revealed that
about 77% of the area suitable for the species is likely to be ~~overflowed~~ within the critical
height range (Fig. 3f). This area, ranging from 278 to 4502 m a.s.l., represents 30.6%
(7871 km², Table 3) of the overall extension of the Swiss Alpine massif.

41 42 345 3.3 Relative contribution of different variables

The environmental conditions that mainly drove the probability of a bearded vulture
flying within the critical height range were steepness of the terrain and food availability
(permutation importance of 29.6 and 19.7%, respectively) (Fig. 1). Bearded vultures
were more likely to fly at lower height (<200 m a.g.l.) not only when approaching steeper
slopes but also in areas with a high probability of ibex presence (Fig. 2), i.e. sectors

where it is more likely to find ibex carcasses. This pattern is evidenced not only by the
PD curves, but also by the increasing concentration of the ICE curves with increasing
values of the two variables. Overall, the probability of flying within the critical height
range was lower over north-facing mountainsides than over south-facing mountainsides
and higher over areas dominated by scree, rocks, and glaciers compared to the
remaining land cover conditions (i.e. forest, anthropic areas, water bodies, etc). Flying
within the critical height range was also more likely to occur in areas typically exposed
to stronger winds compared to areas with weaker winds.

24 25 360 **4. Discussion**

The wildlife vs wind energy conflict model developed here extends commonly applied
approaches of predicting areas of potential collision risk with wind turbine blades based
on mere species' spatial occurrences by adding a vertical dimension. In effect, its
predictions are refined in the sense that it specifically predicts in which areas bearded
vultures would effectively fly within the vertical segment of airspace swept by the blades
of wind turbines if deployed. As the majority of the GPS locations (77.5%) indicated
flight activity of bearded vultures at heights lower than 200 m a.g.l. (Fig. S3), bearded
vultures in the Swiss Alps seem to be active most of the time in the dangerous height
range. This concurs with former findings by Rushworth & Krüger (2014) who estimated
that south African bearded vultures spend 74.7% of their foraging time below 200 m
a.g.l. As a result, our final high-risk conflict map shows that 76.9% of the area of
suitable habitat across the Swiss Alps (i.e. 30.5% of the entire Swiss Alpine range) may
incur some potential collision risk (Fig. 3f, Table 3).

Steep south-facing slopes and areas with high probability of ibex presence, i.e.
providing carcasses potentially exploitable by bearded vultures, offered the best
conditions for low-height flight (Fig. 1-2). Terrain steepness and exposition are indeed
two key factors explaining the formation of updraughts, i.e. both thermals and
orographic uplifts. Thermals are generated by unequal heating of the earth surface:
solar radiation heats up certain land cover types faster than others (e.g. dark rocky
outcrops), thus generating columns of warm air that rise from the ground. Orographic
uplifts stem from the upward deviation of wind blowing against mountainsides. Although
the former occur mostly along sun-exposed rocky slopes and over flat areas [72], the
latter are generated exclusively along slopes and ridges. While thermals in lowlands are
furthermore easily disrupted by dominant winds, mountain regions supply a year-round
source of uplifts, thus providing optimal conditions to support the flight of raptors
exhibiting high wing-loading such as vultures [73]. In line with this, various studies have
showed that raptors fly relatively low over ridges and steep slopes [32,53,74] where
orographic uplift is more likely to occur. As shown here, bearded vultures are no
exception to that rule.

While static and/or dynamic environmental conditions are frequently included to model
flight behaviour in relation to potential collision risk with wind turbines [32,33,52,53,74],
the spatial distribution of potential food supply has – to the best of our knowledge -
never been considered so far. Our model ranked this environmental variable as the
second most important one in explaining the probability of flying within the critical height
range. Vultures, and large soaring birds in general, are constrained in their movements
by the availability of updraughts. By soaring into ascending air currents for gaining

height and then gliding to another location [75], they achieve an energy-efficient
commuting pathway [76,77]. However, the optimal height to be gained with soaring is
likely to result from a trade-off between mobility for long-range horizontal displacement
and the ability to inspect the ground for locating food, the latter diminishing with height
[78]. This might be particularly crucial for bearded vultures which, given their peculiar
bone-based diet [79,80], have to locate carrion and small parts of carcasses that may
be easily overseen. Ibex carcasses represent the most important food supply of Alpine
bearded vultures, to an extent that their distribution is largely driven by the presence of
that ungulate species [40,43]. Our model could even capture that ecological
requirement since the probability of flying within the dangerous height range was always
high over areas with high probability of ibex presence, regardless of the other
environmental conditions (Fig. 2). That a similar pattern was not found for chamois,
another important food source for Alpine bearded vultures, is not surprising because
chamois' distribution is much more uniform as the species is less dependent on a rocky
substrate than ibex.

We used two different validation approaches to assess the ability of our model to
generalise across individuals and geographic regions within the Swiss Alps. In this
respect, the leave-one-bird-out cross validation is important to ensure that model
predictions are not biased by individuals with a prevailing number of GPS bearings.
Only three birds out of 30 (i.e. W361, BG1022, and BG980, Table S3) caused some
marked drop of the AUC when used to evaluate the model. These birds had all been
tagged in France, visiting only occasionally the southwestern most part of the study
area. These notwithstanding, a few birds yielded GPS locations exclusively from one

[revised manuscript text omitted]

^d Topographic position index according to Wilson (1984).

^e Swiss Wind Atlas [48].

**Table 2: GPS-tagged birds included for modelling the flight height of bearded vultures in the Swiss Alps**
 **with mention of the country of first release (or subsequent recapture), origin (C: captive-bred; W: wild-**
 **fledged), year of fledging, sex (M: Male; F: Female; U: Unknown), manufacturer of the transmitter, number**
 **of GPS locations retained after data cleaning (N), total number of tracking days within the Swiss Alpine**
 **range, percent of locations below 200 m a.g.l. (%), and number of GPS locations retained after randomly**
 **subsampling one location per minute (S).**

Bird ID	Country	Origin	Year	Sex	Manufacturer	N	Days	%	S
BG1071	CH	C	2020	F	e-obs	4,855	102	82.92	1,667
BG1068	CH	C	2020	M	e-obs	17,036	85	92.14	1,948
BG1003	CH	C	2018	F	e-obs	831,178	664	84.00	24,356
BG1001	CH	C	2018	M	e-obs	537,155	591	77.22	18,244
BG964	CH	C	2017	M	e-obs	959,434	689	74.83	27,357
BG899	CH	C	2016	M	Microwave	40,771	600	75.95	39,961
BG900	CH	C	2016	M	Microwave	17,330	292	73.04	17,006
BG841	CH	C	2015	F	Microwave	7,263	372	69.95	7,113
BG838	CH	C	2015	F	e-obs	1,386	339	80.38	1,358
BG802	CH	C	2014	M	Microwave	39,759	1,308	73.20	39,112
BG797	CH	C	2014	M	Microwave	12,302	985	74.69	12,162
BG321*	CH	C	1999	F	Ornitela	3,764	263	58.95	671
BG1031	FR	C	2019	F	Ornitela	3,358	32	58.87	421
BG1022	FR	C	2019	M	Ornitela	2,782	1	70.13	77
BG980	FR	C	2018	M	Ornitela	9,779	26	63.43	472
BG983	FR	C	2018	M	Ornitela	5,789	13	72.62	268
BG905	FR	C	2016	M	e-obs	4,001	5	73.41	255
W361	FR	W	2020	U	Ornitela	4,710	19	67.56	303
W356	FR	W	2020	U	Ornitela	170	44	84.71	170
W346	FR	W	2020	U	Ornitela	20,172	15	81.60	666
W285	FR	W	2019	F	Ornitela	139,526	173	74.28	4,734
W284	FR	W	2019	F	Ornitela	29,788	58	60.37	1,306
W313	FR	W	2019	F	Ornitela	11,563	10	68.36	345
W251	FR	W	2018	M	Ornitela	14,482	109	61.89	840
W209	FR	W	2017	M	Ornitela	99,304	137	71.68	3,594
W196	FR	W	2016	F	Ornitela	46,655	986	83.41	12,056
BG998	AT	C	2018	M	Ornitela	44,431	277	76.93	3,152
BG843	AT	C	2015	M	e-obs	31,206	311	58.60	7,214
BG840	AT	C	2015	M	e-obs	407	76	74.55	407
W349	IT	W	2020	F	Ornitela	78	13	75.64	77

* Tagged as adult bird in 2017

**813** **Table 3:** Percentage of shares of the different levels of sensitivity (increasing from 1 to 4) predicted in the
**814** potential conflict map (Fig. 3e) and the high risk conflict map (i.e. where the bearded vulture is likely to fly
**815** within the critical height range, i.e. below 200 m a.g.l., Fig. 3f) in the whole Swiss Alps. The last column
**816** reports the shares of areas where the bearded vulture is likely to fly above the critical height within the
**817** habitat.

Level of sensitivity	Potential conflict map	High risk conflict map	Remaining
1	12.3	8.2	4.1
2	11.0	8.5	2.5
3	7.2	5.8	1.3
4	9.2	8.0	1.2
Total	39.7	30.6	9.2

**818**

**Figure 1** Permutation importance of the environmental variables used to model the probability of bearded
vultures flying below 200 m a.g.l. Permutation importance is presented as the drop in training AUC (%)
when randomly permuting the values of the respective variable within their empirical range. Variable
abbreviations are given in Table 1.

**Figure 2** Marginal effect of the five most important environmental variables for predicting the probability of
 a bearded vulture flying below 200 m a.g.l. In grey are plotted 1000 randomly sampled individual
 conditional expectation (ICE) curves [66] and in blue the partial dependence (PD) curve [67]. For land
 cover, a categorical variable, each boxplot shows the ICE values without outliers and the blue dot the
 value of the PD. The curves for the remaining environmental variables are shown in Fig. S4.

**Figure 3** Predictions of bearded vulture occurrence and the probability of the species flying within the
 risky height range (< 200m a.g.l.) that potentially generate risks of collisions with the blades of wind
 turbines in the Swiss Alps. The maps show a) the predicted probability of a bearded vulture flying below
 200 m a.g.l. calculated as the mean prediction of a 30-bagging procedure and extrapolated to the whole
 Swiss Alpine range (gradient from blue, zero probability; to red, high probability); b) the probability of
 bearded vulture occurrence described in Vignali et al. (2021, Fig. 4e); c) joint probability of occurrence
 and flying below 200m a.g.l., calculated as the product of map a and b. The map showed in d) is the
 translation of a) into a binary response using the threshold for which 95% of the locations occurring at
 risky heights are correctly predicted (the areas with a high probability that a bearded vulture flies within
 the critical height range are shown in red); e) the “potential conflict map” described in Vignali et al. (2021,
 Fig. 4f); f) the “high-risk conflict map” calculated as the product of d) and e).

Appendix B

Manuscript Reference Number: RSOS-211041

Rebuttal Letter

Dear Prof. Enrico Bertuzzo,

Thank you for having handled the review process and for inviting us to submit a revised version of our manuscript entitled “A predictive flight-altitude model for avoiding future conflicts between an emblematic raptor and wind energy development in the Swiss Alps” to *Royal Society Open Science*. In the revised manuscript we have carefully considered all the reviewers’ suggestions and comments. We hope that these changes together with the answers we provide below adequately address all the points you and the reviewers have raised. We also thank the reviewers for their commitment devoted to the revision of our manuscript. Their constructive feedback on our initial submission significantly contributed to improve the overall quality of the article.

Below is a point-by-point response (in red) to the questions and suggestions (in black) provided by the reviewers (page and line numbers refer to the document with track changes).

Sincerely,

Sergio Vignali, on behalf of all co-authors.

Associate Editor Comments to Author (Professor Enrico Bertuzzo):

The manuscript has now been reviewed by two experts in the field. They both find that the manuscript could potentially contribute significantly to the literature. However, they both point at some issues that should be addressed in order to improve the paper and qualify it for publications. They have provided a very detailed and constructive list which I invite the authors to take in full considerations. In particular I would like to further stress few points:

- better describe the methods used and justify it (see also autocorrelation issue).
- address the issue related to the different number of datapoints per bird.
- biological vs political boundary: address it with a new analysis or otherwise justify it explicitly in the main text.

Thank you, these are surely valid points, we addressed as follows: We have explained our choices with regard to the methodology. Moreover, we rerun our analysis based on a modified filtering procedure, which addresses the concerns of the reviewers. Specifically, we addressed the point that with our previous method for filtering out locations of flying birds we might have excluded some fixes recorded at higher altitudes. We also ensured now that the minimum sampling rate is not less than one minute and included only birds with more than 100 data points.

Using this new dataset, we reran the analysis. Results and discussion were updated to accommodate the new analysis. Overall, there are no substantial differences in the results, except

for the relative importance of the variable “ibex probability of presence” which moved from the second to the fourth position in the ranking.

Reviewer 1 stressed that our manuscript would benefit from proofreading. Although all his/her suggestions have been accommodated and many others from reviewer 2 certainly improved the use of English, a native English-speaker colleague proofread the whole manuscript.

Reviewer comments to Author:

Reviewer: 1

Comments to the Author(s)

This paper builds a model from GPS tracking data from bearded vultures to predict the likelihood of a vulture flying within the rotor swept area of a turbine (<200 m agl), based on relevant topographic and environmental variables. Authors use this model to predict the probability of flight <200 m agl across the species range in the Swiss Alps. They then use a previously published predictive map of the probability of bearded vulture occurrence in the same region (Vignali et al 2021) to combine the probability of occurrence and flying in the risky zone to produce a high-risk conflict map to guide sustainable development of wind energy away from areas expected to be highly risky to this species.

I must first note in my review that I am not familiar with the modelling method - deep feedforward neural networks and thus have not commented on the section on the modelling approach. However, the model appear to have been well-cross validated, with both between birds and between region performance being tested. It is hard to tell without being familiar with the deep feedforward neural network procedures, but I would usually expect cross-validation methods to be used to inform model selection in addition to model performance, which has not been done and might be an avenue to explore.

Deep learning has gained popularity only recently and is becoming more and more used in ecology (we have included some references, see p. 12; lines 246-148). It expresses its full potentials when trained with large datasets, from which it can learn complex patterns without having to define a data model. With this large datasets it is not essential to run cross validation for model selection. Moreover, strictly speaking, what we have done is hyperparameter tuning rather than model selection. It is a fine tuning to adjust the values of the hyperparameters to optimise model performance and is typically performed without cross validation for large datasets. Please, see also the following reference cited in the methods section:

Browning E, Bolton M, Owen E, Shoji A, Guilford T, Freeman R. Predicting animal behaviour using deep learning: GPS data alone accurately predict diving in seabirds. *Methods Ecol Evol.* 2018;9:681–692. <https://doi.org/10.1111/2041-210X.12926>

Overall, the paper addresses an important and rapidly growing conservation dilemma and provides maps which will be useful for planning new wind energy developments in locations which pose the least collision risk to this species. This is not the first collision risk map for

bearded vultures, which has already been achieved by Reid et al. 2015 [Using spatial analyses of bearded vulture movements in southern Africa to inform wind turbine placement, *Journal of Applied Ecology*]. Although referenced in a couple of places, there has not been much mention of this work, and a discussion on the pros/cons of the two methods or how this work is different and/or progresses conservation further would be merited.

This is a good point. The main differences are the statistical methods used to model both, distribution and flight altitude behaviour, and the use of natural food supply as explanatory variable of flight altitude behaviour. The statistical methods are extensively explained in our previous work, which also accounted for potential future range expansions of the species, and in this article. Especially the use of a framework suitable for very large datasets (completely reproducible and provided along with the article) could potentially facilitate the analysis of similar datasets. Although we acknowledged that Reid et al. 2015 considered predictable sources of food in their analysis (see below), we have not explicitly discussed the pros and cons of the two methods as it would have extended too much the manuscript. However, if the editor believes this is important, we will surely include it in the discussion.

My main concern is that the majority of the tracking data has come from juvenile, immature and subadult birds. According to Kruger et al. (2014) [Differential range use between age classes of southern African bearded vultures, *PlosOne*] bearded vultures only reach adulthood at 6 years old, thus only one individual was tracked as an adult in this study. Therefore, I think the authors can only really achieve a conflict map for the sub-adult sector of the population. Further more specific comments are given below.

This is surely an important point. Please see our response below.

Statistical methods

I am unable to comment in depth on the primary methodology which I am unfamiliar with – deep feedforward neural network models. It isn't clear to me if different variable combinations have been tested, or if they have all been assumed to be important predictors of flight height. For example, the permutation importance shows that 'eastness' has an importance of around 3%. Has it been tested if the model would be more parsimonious without this variable? Or is this accounted for in the neural network?

We have used the environmental variables that we considered relevant for the flight altitude behaviour of the species. Model parsimony becomes important when the model is projected outside of the area in which data are collected, which is not our case. Compared to classical statistical methods for which a data model should be defined, in machine learning it is the model itself (the neural network in our case) that learns the complex relationships between the variables. Since ten variables are not many for a neural network – and given the extensive amount of data, we did not explore the effect of removing variables.

L153. "average wind speed at 100 m a.g.l. was extracted from Swiss Wind Atlas". Is this the average across the year? Derived from monthly, daily or hourly means? Can you give more information on how it was calculated and the spatial resolution of this variable? Do you think the

average conditions represent the actual conditions available? E.g. an area might be very windy in one season or time of day, but not windy at all at another time. The ‘average’ for such an area would not actually capture either of the conditions available. If this is the case, I would recommend making predictions for actual wind scenarios that occur through the year/day and then averaging the predicted value for each location. This would also require associating each fix with the actual conditions when it was recorded for the analyses stage.

The Swiss Wind Atlas is based on a wind model which provides estimates of annual wind speed with a spatial resolution of 100 m. This model is based on a long time series of climate data measurements spread over the whole country. The specifications are given in the cited document and are probably too detailed to be included in the methods section, but we have included the resolution of the raster layer (p. 8, lines 172-173). This model has been produced for wind energy development to identify broad areas where wind is available. We agree that using actual wind speed information would be more informative when modelling the flight altitude at a given point of time. However, the only way to do this would have been using measurements from the weather stations that are closest to the flight locations, which – however – might not represent the actual wind conditions at the flight location, as wind conditions may be significantly altered by topographical conditions. The modelled and interpolated wind map accounts for this aspect. In addition, we expect that areas with stronger mean annual wind speed are generally relatively windier regardless of wind direction. Moreover, as the Wind Atlas is the main source of information for wind energy companies, these areas are likely to be the ones to which wind companies look for new installations. We therefore considered it important to investigate the relationship between this variable and the flight altitude behaviour.

Age classes and data

Previous work by the authors which this work relies on (Vignali et al 2021), i.e. the habitat suitability model, was modelled separately for juveniles and adults, as well as summer and winter. It seems that the majority of the tracking data used in this paper comes from juvenile or subadult vultures, with only one adult being tagged and one additional bird making it to 3.5 years, and two others to 2.7 years (this was according to tracking days in table 2 – however, this is possibly 9 individuals collected data >3 years old, see below, but it is unclear which individuals or how much data). Nevertheless, following Kruger et al. (2014) bearded vultures only reach maturity at 6 years. Thus, the work here really can only assess juvenile/subadult risk and this should be made clear. As a result, I would recommend removing the limited adult tracking data from the analyses. Furthermore, the habitat suitability map used in the analyses should be an aggregation of Fig 4a and Fig 4c (from Vignali et al 2021), i.e. also not including adult data.

L320 – “Some devices (N=9) recorded data for more than 3 years. This is approximately the age at which juvenile bearded vultures change their behaviour, shifting from an exploring phase to a phase of territory establishment [40,43].”

From Table 2, only one individual had more than 1095 tracking days in the analyses. However, I understand this might be number of unique tracking days in the analyses – not days/age since tagging, which would also be useful in Table 2. Kruger et al 2014, used the following age

classes: juvenile (post-fledging to 2 years), immature (2–4 years), sub-adult (4–6 years) and adult (.6 years), which is quite a big discrepancy from this paper.

You have raised an important point here. We agree that from Table 2 was not possible to determine the age of the birds during tracking time within the Swiss Alpine range. We have updated Table 2 which now includes the date of the first and last collected GPS locations (p. 42) and makes it now possible to reconstruct the age class of the bird. In our previous work (Vignali et al, 2021) we used two age classes: juveniles/immatures including all the birds up to 3 years and subadults/adults including all older individuals. Considering this classification, 10 birds contributed data after reaching the subadult/adult class. This justifies using the habitat suitability map showed in Fig. 4f of our previous work (i.e the one which includes both age classes). Moreover, the shape of wings and tail change gradually, so there is no prominent morphological difference between subadults and adults as there is between juveniles and adults. We believe that the 2 adults should be included in the model as they potentially bring additional information to the network. However, we agree that adult birds are underrepresented, and we address this issue in the discussion (pp. 25, lines 548-558). Also the sentence you quoted as been moved from the results to the discussion (p. 16, lines 346-349).

Also, how do you think having different ages, from fledgling to subadult might have affected the results?

We believe that modelling the age classes separately and then merging the predictions would not change the overall prediction of our model. In line with the aim of our study to have a general model for the species rather than investigating differences between age classes we decided to merge the data of different ages.

L307-310 and Table 2- One individual had only one day of tracking. Is it worth keeping this there? What happened to this individual? Is the tracking data likely to represent ‘normal’ behavior? Further, I see this individual is one of the ones which had lower AUC cross-validation score (L416).

Thank you for pointing this out: one tracking day does not mean that this bird provided only data on one day and disappeared or died afterwards. Birds were released in four different countries: Austria, France, Switzerland, and Italy. Information on the country of release was available in Table 2 (p. 41) but we recognise that it was not sufficiently clear and we have added a sentence in the text (p. 9, lines 181-183). This bird was released in France and visited only for one day the Swiss Alpine range, so there is no reason to consider its behaviour anomalous. However, in line with the suggestion of reviewer 2, we have now only included birds that collected at least 100 GPS fixes (p. 11, lines 231-232), thus this bird is not anymore included in the new analysis.

Other

L361-373. Intrigued as to why these results haven’t been compared with Reid at el 2015 – who found a total of 66% on non-adult bearded vulture flight <200 m agl.

We apologise for this. This is now acknowledged (p 22, lines 474-476).

L369. Rushworth and Kruger 2014 not in reference list.

Thank you for reporting this, now it is included (p. 31, lines 698-699).

L382 “the spatial distribution of potential food supply has – to the best of our knowledge - never been considered so far”. Although in a slightly different format, Reid et al 2015 included the variable “distance to vulture restaurant” which is equal to the distribution to reliable food sources for bearded vultures.

We consider the sentence is still valid since the distribution of natural food sources is not the same as distance to artificial feeding sites. However, we have modified the sentence and mention the study of Reid (p. 20, lines 443-444).

Minor

L89 Suggest removing “radio-” since it doesn’t have the fine precisions that GPS tracking can achieve; particularly not altitude measurements.

Right, we have removed it, (p. 5 line 95).

L.155 I am surprised TPI and slope unevenness are not correlated – what was their correlation coefficient?

It was 0.014 according to 10,000 random samples and the Spearman’s correlation coefficient.

L201. Since you have a bunch of different tags which might record speed slightly differently, I would suggest using a between peaks minimum as a unique cutoff for each individual – following Schaub, et al 2020 [Collision risk of Montagu's Harriers Circus pygargus with wind turbines derived from high-resolution GPS tracking. Ibis]

Thank you for your suggestion. Authors used different methods to filter out locations according to the ground speed. We agree with you that we should have done it differently. Following the suggestion of reviewer 2, we now used a combination of instantaneous ground speed and flight altitude (p. 12, lines 222-223). Please see our answer to reviewer 2 for more details. However, after remodelling the flight altitude behaviour using the dataset culled with this new criterion the results did not change substantially.

L208 – one location per minute seems like autocorrelation may still be an issue?

Please, refer to our answers to reviewer 2.

Very minor. The paper would benefit from English proof reading, some very minor points are given here:

L24 “..with future wind turbine deployments...”. Edit plurals.

Thank you, corrected (p. 2, line 25).

L30 Change “expansions” to “expanses”

Thank you, corrected (p. 2, line 32).

L56. Edit needed to second half of sentence.

Edited, it reads better now (p. 3, lines 58-60).

L87-88. "...as spatially-explicit predictive models allow extrapolation to areas...". Grammar edits.

Ok thanks, corrected (p. 5, line 93).

L92. Change 'gate' to 'door'.

Thanks, door is more appropriate (p. 5, line 98).

L103. suggest changing "new" to "growing".

Ok, we changed with "growing source of hazard", (p. 6, lines 109-110).

L105-106. Suggest restructuring sentence to avoid "if". Perhaps "Using a spatial model...?"

Thank you for the suggestion, we have reframed the sentence (p. 6, lines 112-116).

L151. Changes 'providers' to 'sources'

Thank you, changed (p. 8, line 170).

L225. Comma ',' after 'very large datasets'.

Ok, included (p. 8, line 250).

L287. ..., this regardless of...

Thank you, corrected (p. 14, line 312-313).

L493. "believe"

Right, corrected, thank you (p. 26, line 574).

Reviewer: 2

Comments to the Author(s)

This is a nice paper on risk mapping and it is an advancement over previous risk mapping. The two prior raptor-wind turbine risk maps based on flight behavior (Reid et al. 2015 (citation 33 in the ms) and Miller et al. 2014 (not cited, but I've suggested it in the text) don't do a great job of combining probability of occurrence with probability of risky flight. This paper explicitly does that and so it is a step forward.

There are a number of changes that I think would make this manuscript better. A few of the most important items here:

- I don't agree with your use of the term "flight height". It is vogue to use it, but it is also incorrect. You note Peron's definition of "flight height" (your line 175). I appreciate that he defined this, and I appreciate that you explain how you chose your terms, but his definition leads

to sloppiness. Height describes how tall something is. I've corrected it a few times in the pdf, up to you and the editors for a final decision.

Thank you for clarifying the misuse of "flight height". As you pointed out, "flight altitude" is the correct terminology so we have changed the manuscript accordingly.

- I'm curious as to why you limited your modeling to a political region (Switzerland) rather than a biologically defined region. You have lots of evidence that the birds ignore political boundaries. For your reports to funders, sure, a political region makes sense. But for a biological study, a biological region makes more sense.

We agree with you that considering the biological region makes more sense. However, GIS layers are created differently from the different countries. Spatial resolutions are different and not all required information is available for France, Italy, Austria, and Switzerland. Trying to homogenise these layers would surely introduce a bias. Moreover, crucial information such as food availability would have been missing in the other countries. Since in Switzerland you find all the habitats available in the Alps and the country hosts more than a third of the breeding territories, we consider Switzerland a representative study region for addressing our question. We justify our choice now explicitly in the methods section (p. 7, lines 140-147).

- There needs to be a much more detailed description of your data filtering process and the thresholds you use for inclusion in the analysis. You could do this in the SI (if there are space limitations in the ms) but better in the main ms if you can.

This might have been confusing but all the information concerning the filtering process was in the methods, nothing more was done. Probably the statement "32 individuals fulfilled the requirements for being included in the analysis" was confusing and has been removed (p. 15, lines 332-335). With requirements we meant all filtering process:

- 1) birds equipped with GPS devices which measured flight altitude and instantaneous ground speed were used to select moving locations (there were many GPS tags that didn't record flight altitude).
- 2) birds that collected data within the Swiss Alpine range
- 3) birds that recorded more than 8 weeks (excluded to reduce the bias related to the release event).

In the new analysis we also excluded birds which collected less than 100 fixes (p. 15, lines 231-232) to increase the minimum number of tracking days.

- I am concerned that your data set includes birds with only 1 day of data collection and as few as 77 data points and other birds with as many as 1,300 days and 40,000 data points. Despite this, your modeling approach does not include a random effect for bird, which means that individuals are weighted differently. If you can set a random effect for bird, that would be best. However, since it is probably not trivial to include a random effect in your models, you should probably set some threshold for inclusion and justify it, either with citations or with a simulation exercise.

In the new analysis we have excluded birds with less than 100 fixes (see above) and now the minimum number of tracking days is 5. Machine learning algorithms don't allow accounting for random effects, despite probably Random Forest that recently has been extended to include random effects, see:

Pellagatti, M., Masci, C., Ieva, F., & Paganoni, A. M. (2021). Generalized mixed-effects random forest: A flexible approach to predict university student dropout. *Statistical Analysis and Data Mining: The ASA Data Science Journal*, 14(3), 241-257.

The way how to choose a threshold to exclude birds is quite arbitrary. Birds with only few fixes are those released in the neighbouring countries that visited Switzerland only sporadically (or only for one day as the case of the bird that you mentioned). These birds can still bring information to the network and the potential bias was addressed with the leave one bird out cross validation. However, probably only one day of tracking days was too little and we increased the threshold to 100 fixes (p. 15, lines 231-232).

Based on the leave-one-bird-out cross validation, we do not believe that this changed the results, which are most likely influenced by the new criterion used to select flying birds (several locations collected above 100 m a.g.l. were indeed excluded from the previous analysis).

- I'm concerned about autocorrelation in your dataset. I don't know the modeling techniques you describe well enough to understand if they handle autocorrelation well or not. That said, you randomly selected one obs per minute. That means the data are anywhere from 1s to 120s apart. There is still high autocorrelation in data collected at that time interval. As such, your statistical models need to handle autocorrelation. If you were using a more standard modeling approach, I could include some suggestions – certain non-parametric model types handle correlated data, or imposing a correlation structure on your models works, etc. But I really don't know what to suggest in this case – that one is up to you.

Since we have rerun the analysis to accommodate the new filter for non-flying fixes, we also modified the subsample of the fixes collected at 1 Hz which is now regular at 1 minute resolution and not random anymore (p. 11, line 228; table 2 p. 42; Supplementary material Fig. S6). Artificial neural networks have been used to model data collected at higher resolution than 1 Hz. It is plenty of examples in computer science, applied, for example, to the human activity recognition dataset or in medicine to analyse time series of clinical data. In ecology, the authors of the cited article [64] (in the revised manuscript) resampled at 100 seconds but in [63] authors resampled data at 1 Hz.

We have included another reference [66]:

Dufourq, E., Durbach, I., Hansford, J. P., Hoepfner, A., Ma, H., Bryant, J. V., ... & Turvey, S. T. (2021). Automated detection of Hainan gibbon calls for passive acoustic monitoring. *Remote Sensing in Ecology and Conservation*.

In this article, the authors use deep learning to recognise the presence of gibbons from an acoustic monitoring. They use data at 4800 Hz, with windows of 10 seconds (i.e. 48,000 data points) and with moving window shifts of 1 second. Their architecture of the neural network is

different, to accommodate the nature of their data. Specifically, they used convolutional neural network to extract features from 10 seconds worth data recording. However, after feature extraction, the last two layers are “Dense layers” (see Fig. 4 in the mentioned article), namely the same that we have used in our network.

We also explicitly wrote in the text that artificial neural networks have been used to model data with high temporal resolution (p. 12, lines 247-248).

There are ways to account for autocorrelation in deep learning, namely recurrent neural networks and they could be used to model trajectories, make predictions of movement or behaviour. However, since we had another research question, we chose the method that we thought was best to answer it.

- Several spots in the ms could be substantially shorter.

We deleted some sentences, especially in the introduction (see our response to your comment below). However, after addressing the suggestions of the reviewers, the overall length of the manuscript has not substantially changed.

- The literature cited and citations in text have some issues to address. In most places in the text you use numbers (presumably to match journal format), but in a number of places you use author names. Try to clean that up, as this got confusing in a few spots.

You are right, we used text for the reference in few places, we have replaced them with numbers.

Please see the pdf for some minor comments and some repetition of things I’ve written below ("**RSOS-211041_Proof_hi_review comments.pdf**")

We found your comments on the pdf very useful and we implemented most of them.

Title – I would change “height” to “altitude”

Thank you, we have replaced the word height with altitude throughout the manuscript.

Introduction – In general, I think the introduction is pretty strong. It would be good to shorten it a bit. The last two paragraphs could be combined and shortened a bit – you don’t need to go into the methods too much here, for example. You also have two paragraphs that each only have 2 sentences. Generally it is better to have at least 3 sentences in a paragraph. You can shorten and eliminate some of this text to combine. For example, the last 2 sentences of the first paragraph contain information repeated somewhat in the second sentence of the second paragraph and the sentence that starts at line 69. All that said, the intro is strong and relevant to the topic and mostly well cited.

Thank you for this suggestion. Part of the text was a bit redundant and we have removed the sentence at p.3 lines 49-51. We have also removed the paragraph breaks at p.3 line 51 and p. 4 line 79. We have kept the part related to the methods as we consider it an important part of the introduction.

I’ve put a number of notes in the pdf. A few to expand on here.

- “lift” is a force generated by the wing (the airfoil). So the words “orographic lift” are nonsensical (again, lots of people use this term, but it is incorrect). Simple fix – “orographic updraft”. Also, “thermal updraft” (not “thermal lift”).

Thank you to clarify it, we have change with updraught.

- You suggested that we need planning tools to avoid deploying wind facilities in areas where conflicts with biodiversity will occur. Unfortunately avoiding conflict is a dream we are unlikely to realize (look at projections for expansion of wind in the coming 30 years – there will be lots of conflict). We need to talk about “minimization” of conflict, not avoiding it. Easy but important change to the ms.

You have raised an important point here, for avoiding conflicts it’s probably too late. We have reframed our sentence (p. 4, line 78).

Methods – I’m curious as to why you made the decision to limit the modeling exclusively to Switzerland. I get that the funding is focused on one country. That said, I have a hard time believing that your tagged birds never left the country. Clearly also, the alpine zone extends beyond the borders of this one country. This is mostly splitting hairs and is certainly not a reason to reject the manuscript. But wouldn’t it be more reasonable to model a biologically relevant region, rather than a politically relevant region? Perhaps the “contiguous Alpine areas within 500km of known nests of vultures” or something like that?

My question became more relevant when I realized that many of the birds whose data you used were tagged outside of Switzerland. I think there is good rationale for using a biologically defined study area, rather than an politically defined one.

Please refer to the answer above

Can you go into a bit more detail as to what is an “anthropic area” in Switzerland? In the USA, we use the term “urban” for areas in cities. But from my perspective, 90% or more of Switzerland is dominated by humans. As such, specificity seems really important when clarifying your land cover classifications.

Thank you to point it out, we have realised that this information was not available. Although with Table S1 we provided the reclassification table for the land cover layer, the value of the field used for the classification was in German. Anthropic areas include all areas with human activities (settlements, quarries, and pits) or without a natural cover (paved tracks and dams). We have included a new column in table S1 with the description of the areas included in each land cover type.

Line 170-171 – what is the slowest rate of data collection of any GPS devices?

With the new filtering process is 1 fix per minute, see previous answer.

Rather than simply using flight speed as a threshold to remove “non-flighted” points, you should use both flight speed and altitude. A vulture at high altitudes in a thermal might be moving very quickly upwards but very slowly in a horizontal plane. See both Murgatroyd 2021 and Poessel 2018 for this. Murgatroyd says that Schaub et al. 2020 Ibis also does this.

Thank you for this suggestion. You are right, by using only flight speed we had excluded some GPS locations occurring at higher altitudes. We have implemented what you suggested in the pdf, i.e. we have included fixes collected at speed < 2m/s occurring at altitudes > 100 m a.g.l., and rerun our analysis. Overall results are very similar, despite slight changes in the ranking of the variable importance. We have described the new filtering criterion in the methods (p. 10, lines 222-223) and the results and discussion have been modified to in line with the new analysis.

I'm concerned that some of your birds have < 10 days of data collection and that you are modeling birds with so few days of data collection in the same batch as birds with more than 1000 days of data collection. I think it would be good to set a minimum threshold for inclusion in your models. Maybe 15 or 20 days? Likewise, set a threshold for a minimum number of points. Maybe 150 or 200 data points, or at least 1 per day over your minimum time period?

Thank you, we set the minimum number of data points to 100, please refer to the answer above.

OK - having read the results, you say that you had "requirements for being included in the analysis". I missed those in the methods. Can you clarify in the methods what were the requirements for being included in the analysis? I suspect that will address my concerns above. Likewise, be sure to explain whether your filtering of GPS points was before or after you filtered birds.

Please, refer to the answer above.

I don't love your resampling down to 1m data. The resampling itself is fine. However, the bigger problem is that you don't clearly explain the slowest data rate your transmitters collected. 1m intervals is fine, but if some units collected data at 1hr intervals, I'd feel uncomfortable modeling 1m and 1hr data at the same time.

I also don't think that sampling 1 obs per minute removes autocorrelation in your dataset. Especially for a distributional model. Does your modeling approach address autocorrelation well? If not, then you will need to address this.

This is a valid point, partially answered above. It is not a distributional model, we would rather call it a classification problem. The neural network is trained to predict if a given fix occurs above or below 200 m a.g.l. given the environmental conditions. In this kind of problems, the output is given as probability and a threshold must be defined to decide the class to which the predicted value belongs. Your point is surely relevant when trajectories are analysed. Trajectories collected, for example, at 1 Hz have surely different characteristics from trajectories collected at 1 minute temporal resolution. In this case, in contrast, it would not be possible to merge different datasets together.

To avoid confusion, we have removed the word "autocorrelation" from the text because the reason of subsampling was only to reduce the overrepresentation of data collected at 1 Hz (p. 11, lines 229-230).

I really liked that you combined probability of species occurrence with that of flying within a dangerous altitude. That is a gap that has been missing in many studies.

Thank you. Although the model approach was different, Reid et al, 2015 combined them in the same way. This article was cited in the methods (p. 15, line 320).

I'm concerned that your modeling approach doesn't really seem to have a random effect for bird ID. This is problematic since your sample sizes are so different among birds. Partially this will be fixed when you explain your filtering procedure. That said, if some modeled birds had 77 data points and some had 40,000, then you have a bias issue in your analysis.

Please, refer to the answer above.

I think it would be good to explain your modeling approach a bit more clearly. The approach you took is not typical and neural networks have been critiqued in the biological literature of late (specifically in classification fields – they often perform worse at classification than do other models – the Nathan 2012 paper you cite as an example of neural networks being popular actually shows them performing considerably worse than several other methods considered in that paper; sometimes they actually had the worst performance). So, in brief, try to more clearly explain your modeling approach. Be sure to include a description of the assumptions of the models.

Since 2012 there have been a lot of changes. We have included more citations, please, refer to the answer above.

Results –

Table 2 – it would be really good to include the sampling interval – max and min time between GPS fixes, for each bird. That is kind of important to note, since you combine all of them together.

Thank you for this suggestion. We have included the fix rate given as median (min – max) rate. See the new Table 2 (p. 42) which includes also additional information requested by reviewer 1.

I think your opening paragraph of this section can be shorter and more efficient. Try to cut the length by 33-50% but keep the same amount of information. For example, the first few sentences (lines 307-313) could be replaced with the following:

“After filtering, we retained data from 30 bearded vultures tagged in Switzerland, France, Italy or Austria and tracked from Sept 2014 to Dec 2020.”

Everything else should either be in the methods or is repetitive or not needed.

You really do need to explain your filtering approach in the Methods section.

We agree with you, it reads much better (pp. 15-16, lines 335-339). We also explain the criterion used to include a bird (i.e. a minimum of 100 GPS fixes) in the methods section (p. 11, lines 231-232).

As noted previously, I'm not comfortable with including birds with only 1 day of tracking data – unless handled very carefully, you are going to weight those birds disproportionately.

We have included in the new analysis only birds that had at least 100 GPS fixes. The minimum number of tracking days is 5 now.

In your results, it is really important to present the figures in order of appearance. Right now Fig 3 is referenced before Fig 1 and 2. I suspect you switched things around at some point but it is confusing.

As presently shown, the risk map is not the “final” product. To me that is the most important component of your results, so I’d rather see that last.

Thank you, we agree and we have rearranged the text moving the sentence (p. 18, lines 390-394) and changing the headers of the sub sections (p. 16, line 355 and p. 17, line 372). Please notice also that elevation range and area are slightly different due to the new dataset derived from the new filtering process.

Discussion – your first sentence is only partly correct. There are models that include a vertical dimension, there are none that I’m aware of that include both distributional models and flight behavior. That is an improvement, you should be more clear about this.

This is exactly what we meant to say with our first sentence, we specified it by adding “of flight behaviour” (p. 18, lines 399-400).

In general, I found the discussion to be a little underwhelming. To me, your most important result is the risk map. You should start with that and discuss what it means. Two this to discuss are the improvements you make to prior risk modeling and the relevance to vulture conservation. You discuss both of these but not really in a manner that is organized to highlight the really cool aspects of your work. I suggest reorganizing and trying to focus more on key messages and meanings.

We rearranged some paragraphs but did not further expand the discussion, but see below.

You make a compelling argument in the ms that bearded vulture distribution depends strongly on the distribution of ibex. That is pretty neat but I think it is also atypical for many soaring species that interact with wind turbines at this spatial scale. Golden eagles, griffon vultures, white-tailed sea eagles, bald eagles, etc. – all of these tend to be broadly distributed across the landscape and updraft (or water) is often more important than prey distributions for these species. You might want to add a sentence or two talking about how differences in distribution might influence similar risk maps for other species. Since this is such a novel part of your work, talking about this would highlight your novelty and also characterize the general relevance to other taxa.

Ibex limbs are probably the main food source for bearded vultures in the Swiss Alps but in other areas domestic cattle are more important. Other species are more broadly distributed but often their habitat use is affected by food resources. Investigating the effect of the distribution of food supply on flight altitude behaviour is certainly valuable. We have emphasised this in the discussion (p. 21, lines 463-469).

Tables and Figures – Table 2 needs to include the fix rate (time between fixes).

Thank you, we included this information (p. 42), see your previous comment.

Figure 3 confused me. You have three different color scales and apparently 3 different types of information included. After reading the caption several times, I think I understood it. I think you

can fix the caption by making clear that the figures show the progression of your research approach. This is not just a data figure, it is a graphical representation of your approach to analysis. Say that in your first sentence. Perhaps something like “Graphical representation of the research approach used to model risk to bearded vultures from wind turbines. Maps show the data layers used as input into models and the risk maps that are the final product of that modeling exercise.” Also, your color legends need more explanation. Simply showing classes or a range from 0 to 1 doesn’t help much.

This is a very good suggestion. We have included the suggested sentence, partially modified because the layers are not an input of the models. We also tried to improve the description of the colour legends (p. 49, lines 957-971).

I think this is a really nice paper that represents an improvement on past work. With a bit of effort, it will make a very useful contribution.

Appendix C

Manuscript Reference Number: RSOS-211041.R1

Rebuttal Letter

Dear Prof. Enrico Bertuzzo,

Thank you for having handled the review process and for inviting us to submit a revised version of our manuscript entitled “A predictive flight-altitude model for avoiding future conflicts between an emblematic raptor and wind energy development in the Swiss Alps” to *Royal Society Open Science*. In the revised manuscript we have carefully considered all the reviewer’s suggestions and comments. We hope that these changes together with the answers we provide below adequately address all the points the reviewer has raised.

Below is a point-by-point response (in red) to the questions and suggestions (in black) provided by the reviewers (page and line numbers refer to the document with track changes).

Sincerely,

Sergio Vignali, on behalf of all co-authors.

Associate Editor Comments to Author (Professor Enrico Bertuzzo):

Comments to the Author:

The manuscript was reviewed by Reviewer #2 who initially suggested a major revision and, as the authors can read, they are now satisfied with the revision. I carefully checked how the authors responded to the comments raised by reviewer #1 and also in this case I do not see any further revision required.

The manuscript can therefore be accepted for publication pending a minor revision to address the residual minor comments from the Reviewer.

Reviewer comments to Author:

Reviewer: 2

Comments to the Author(s)

Thanks for the careful reply to the reviewer comments. I have many fewer suggestions on this version of the manuscript.

Two brief comments on your response to reviewers.

First, in response to a comment from reviewer 1 about wind measurements at a specific site – you say “However, the only way to do this would have been using measurements from the weather stations that are closest to the flight locations...”. That is not quite correct. There are R packages (RNCEP) and web applications (MoveBank) that can use modeled data to interpolate

to a GPS location and date/time. They are flawed in the same way as you note for weather stations, but they are available. The more pertinent response to this comment though is that by using an “average” dataset, you enable prediction much more effectively. It isn’t much good to predict to specific wind conditions, since you never know when those specific conditions will occur. Much better to predict based on an average, then you can understand the future better. I don’t think you need to change anything, I bring this up mostly for clarification.

Thank you for this clarification.

Second, in my first review I noted that there needed to be a better description of the filtering process. You then went into a bit of detail on filtering for birds. I fear that I was not clear in my description of what I meant by filtering. Presumably you also did some filtering for GPS errors. For example, it is not uncommon to see 3 GPS fixes at 100m in altitude, then a 4th at 2000m, then three more at 100m. In that case the one in the middle is probably a GPS error – especially if time intervals are short. We see this frequently, no matter who the manufacturer. There are also locational errors occasionally – for example, a colleague in Australia sometimes gets GPS fixes in Africa. One out of every 1,000 or 10,000 GPS data points, but it happens. Did you do any filtering for this type of GPS error?

Sorry, we have misinterpreted your point. However, all the filtering procedure is described in the methods. We filtered out fixes with high position error (using the HDOP or the error estimate provided by the manufacturer) (pp. 9-10, lines 196-199) and removed fixes recorded at unrealistic altitudes (p. 10, lines 199-202). Many bursts collected at 1 Hz have been visually inspected and we did not notice such a pattern (both, altitudinal and horizontal displacement) but we did not systematically check for it (probably it occurs more when the fix rate is coarser than 1 Hz). This said, we believe that this is not a problem for the model as Artificial Neural Networks are quite robust to noise. Moreover, as you pointed out, this problem can be detected only for data collected at high temporal resolution and accounting for it could maybe introduce another bias given that some devices collected data at coarser resolution. Extreme cases of horizontal errors, as in your example, are surely not included in the analysis as we filtered out all the locations recorded outside of the Swiss Alpine range.

Manuscript –

Your abstract is mostly introduction and methods. You have only one sentence of “results”. Generally, it is better if an abstract is about 25% introduction, 15-25% methods, 40% results, and 10-20% discussion/conclusions. It would be good to insert more results into your abstract.

We reframed part of the abstract including one more sentence for the results. There is a limit of 200 words (there are 199 now) and it reads well so we would prefer not to change it any further (p 2).

Introduction –

Line 52 – remove “endangered or threatened” and replace with “of conservation concern”. Endangered and threatened have very specific (sometimes legal) meanings, and we are often concerned about species that don’t meet that threshold.

Good point, thank you for this suggestion (p. 3, line 54).

The introduction is very nice – concise, to the point.

Methods –

It would be good to clarify the data filtering you did, as I note above.

See the answer above.

You should clarify the inter-fix interval for the transmitters. Not only how many fixes per day (which you report in Table 2, I think), but also the time between fixes (minimum and maximum for each unit, this can also be in Table 2).

Please, see the answer below for the first question in Results.

I like the explanation for modeling just in Switzerland, thank you.

This information was missing, so thank you for pointing it out in the previous revision.

Line 155 – I would change “bones” to “food” – bearded vultures will eat meat too, if available.

Ok changed (p. 8, line 157).

Line 183 – instead of “Poessel et al”, should this be (53)?

Yes, changed, thank you (p 9, line 185).

Line 205 – since this is an “and”, there really isn’t a “Second criterion”, right? It is just one criteria with two parts? A pedantic point, but you might rephrase to say “Thresholding in this manner may have removed some valid flying positions....”

Ok, we reframed the sentence according to your suggestion (p. 10, lines 207-208).

Line 176 and 210 – If I read this correctly, for the telemetry units that collected data at long time intervals (more than 1 minute), you did not resample to 1 minute, you just used the data as they came, correct? So data analyzed were as fast as 1 fix/minute and as slow as about 1 fix every 15 minutes, correct? You should clarify this for the reader.

This is in the text “and sampled one observation per minute in the case of bursts collected at 1 Hz resolution” (p. 10, lines 212-213).

Line 227 – change “it has” to “they have”

Right, changed (p.11, line 230).

Line 233 – change “heights in” to “heights of”

Corrected, thank you (p. 11, line 236).

Line 237 – is a 0 for a location outside of the critical altitude range? If so, perhaps say that.

Yes, it is 0, we have written it explicitly now (p. 11, line 241).

Line 252 – delete extra space between “30” and “%”

Ok, removed (p. 12, line 255).

Line 255 – is “epochs” the correct word here? I know what it means, but I am not familiar with its use in this context. (also used in line 284)

Yes, it is the right terminology.

Line 265 – perhaps change to “we used 10 km blocks, to verify the ability....”

Thank you, it reads better with this change (p. 13, line 268).

Results –

Table 2 – this might be better as an SI table. Also, “fix rate” needs to be a rate, not a count. Is it “per day”? or something else? Can you also report the “inter-fix interval” – which is sometimes 1Hz and sometimes a lot slower?

This information is already in Table 2. Probably “fix rate” was a wrong terminology and it caused confusion. We have renamed it with “inter-fix interval” and provided the unit (seconds), it should be clearer now (pp). We did not move Table 2 to the supplementary material but if the editor thinks this is necessary we will do it.

Line 324 – “after applying the subsampling procedure” – this is fine but you don’t call it “the subsampling procedure” in the methods. This is the first time you’ve used this terminology. You should name it in the methods and then use that name here. Minor point, easy to fix.

Ok, thank you, we removed “subsampling procedure” to avoid confusion (p. 39, lines 900-901 and p. 40).

Line 345 – 347 – this is interesting, as wind speed is often correlated with flight altitude. It is not a linear relationship though – at low wind speeds, birds are at low altitudes, at moderate wind speeds bird are at higher altitudes, and at very high wind speeds, birds are at low altitudes. Lanzone et al. 2012 shows some of this, as does Bonner et al. 2021 (online early). Does your modeling approach allow for non-linear relationships in your data?

Lanzone, M., T. Miller, P. Turk, D. Brandes, C. Halverson, C. Maisonneuve, J. Tremblay, J. Cooper, K. O’Malley, R. Brooks, & T. Katzner. 2012. Flight responses by a migratory soaring raptor to changing meteorological conditions. *Biology Letters*, 8:710-713.

Bonner, S.R., S.A. Poessel, J.C. Brandt, M.T. Astell, J.R. Belthoff, and T.E. Katzner. Drivers of flight performance of California condors. *Journal of Raptor Research*. In press.

Nothing for you to change here, but you may want to explore this in the discussion.

Yes, the modelling approach accounts for complex non-linear relationships (see pp. 23-24, lines 513-515). This is surely an interesting point but we did not further elaborate on it as the discussion is already quite long.

Figure 2 – your axis labels are inconsistent. The plot for slope does not have 1.00, the others do.

Thank you, we did not notice this, now they are consistent and span from 0 to 1 (p. 44). We also applied the same change to Figure S4 in the supplementary material. Moreover, we updated the R script in Dryad to reflect these changes in the code.

Figure 3 – I really like this one and the new caption is helpful. Can you make the country borders black instead of grey? They are hard to see. Also, since your discussion is mostly about the “Swiss Alpine Range”, you probably should clarify where that is on the map.

We used dark grey and they and the borders are much more visible now, thank you for the suggestion. We also plotted the Swiss Alpine range below maps e and f (it was already below d) and made the border darker (also explained in the caption) (p.46, lines 934-935). We also applied the same changes to Figure S5 in the supplementary material.

Discussion –

Line 370 – also not surprising since steep slopes generate orographic updraft. Orographic updraft tends to subside at about 200m above ground, and so birds can't get high. When they use thermals, they can get much higher. See Katzner et al. 2012 for an explanation of this (citation 55 in the ms). (ok, you get into this a bit in line 393, although you don't explicitly say that orographic updrafts keep birds at lower altitudes, you may want to say this).

Thank you for this suggestion, it make the whole point clearer (p. 18, lines 396-398).

Line 383 – change “exposition” to “exposure”, also change the end of the sentence to “explaining the formation of thermal and orographic updraft”

Right, thank you, we changed it (p.46 , line 387).

Your AUC is 0.7, you might also want to report the “lift” – some metric that compares your predictions to random. I think your lift is about 30%, but not sure.

AUC already compares to random, a random guess would have an AUC value of 0.5.

Line 480 – 483 – not sure what this sentence is trying to say. Perhaps rephrase this for clarity.

We slightly changed the sentence, hopefully it is clearer now (p. 22, lines 486-489).

Line 483 – 485 – perhaps rewrite this entire sentence, “Although we adopted a conservative approach, our results suggest that birds fly above the critical altitude range in about 9.1% (2,351 km²) of the areas previously classified as having high conflict potential.”

Thank you, it reads much better, we adopted your suggestion (pp. 22-23, lines 489-493).

Line 486 – 496 – I'm not worried about the age issues, as was the other reviewer. In my experience, if a bird is in a certain spot, it will respond to the environment in a set way, regardless of age. What differs about flight behavior is ranging behavior - things like home range size and degree of wandering. But flight altitude is fairly constant, regardless of age. Citations for soaring birds that confirm this are:

This paper shows age- and season-related difference in ranging behavior:

Miller, T.A., R.P. Brooks, M.J. Lanzone, J. Cooper, K. O'Malley, D. Brandes, A. Duerr & T.E. Katzner. 2017. Space use and home range characteristics of Golden Eagles (*Aquila chrysaetos*) in eastern North America during breeding season and winter. *The Condor*. 119: 697-719.

These two papers show no age effect for flight behavior:

Lanzone et al. 2012 (as cited above)

Katzner, T.E., P.J. Turk, A.E. Duerr, T.A. Miller, M.J. Lanzone, J.L. Cooper, D. Brandes, J.A. Tremblay & J. Lemaître. 2015. Use of multiple modes of flight subsidy by a soaring terrestrial bird, the golden eagle *Aquila chrysaetos*, when on migration. *Journal of the Royal Society Interface*. 12: 20150530.

This is a good point and we agree with you, although generally a different behaviour might occur between immature unexperienced birds and adults. But we are also confident that this is not a point of concern in our case. We have partially reframed our statement and included the above references to accommodate this perspective (p. 23, lines 502-508).

Lines 434 – 510 – to me, this section is a bit long and it doesn't really get to the biology. It is methodological and focuses a little too much on weaknesses. Every paper has weaknesses, science is incremental, and each new iteration improves on the prior one. That is normal. I think it is fine to mention these things and you can definitely leave this as it is. However, if the editors tell you to make the paper shorter, this is the section you should shorten. You can probably cut it in half, while bringing up the same points and making your paper more impactful.

There was no request from the editor to shorten the manuscript so we did not revise this part, but thank you for your suggestion.

Nice job with this paper, I think it will be influential.

Thank you, the manuscript improved thanks to your valuable comments.